# Recent Progress on Tailoring the Biomass-Derived Cellulose Hybrid Composite Photocatalysts

**DOI:** 10.3390/polym14235244

**Published:** 2022-12-01

**Authors:** Yi Ding Chai, Yean Ling Pang, Steven Lim, Woon Chan Chong, Chin Wei Lai, Ahmad Zuhairi Abdullah

**Affiliations:** 1Department of Chemical Engineering, Lee Kong Chian Faculty of Engineering and Science, Universiti Tunku Abdul Rahman, Kajang 43000, Malaysia; 2Centre for Photonics and Advanced Materials Research, Universiti Tunku Abdul Rahman, Kajang 43000, Malaysia; 3Nanotechnology & Catalysis Research Centre (NANOCAT), Institute for Advanced Studies, University of Malaya, Kuala Lumpur 50603, Malaysia; 4School of Chemical Engineering, Universiti Sains Malaysia, Nibong Tebal 14300, Malaysia

**Keywords:** biomass, cellulose, photocatalysts, hybrid materials, degradation performance

## Abstract

Biomass-derived cellulose hybrid composite materials are promising for application in the field of photocatalysis due to their excellent properties. The excellent properties between biomass-derived cellulose and photocatalyst materials was induced by biocompatibility and high hydrophilicity of the cellulose components. Biomass-derived cellulose exhibited huge amount of electron-rich hydroxyl group which could promote superior interaction with the photocatalyst. Hence, the original sources and types of cellulose, synthesizing methods, and fabrication cellulose composites together with applications are reviewed in this paper. Different types of biomasses such as biochar, activated carbon (AC), cellulose, chitosan, and chitin were discussed. Cellulose is categorized as plant cellulose, bacterial cellulose, algae cellulose, and tunicate cellulose. The extraction and purification steps of cellulose were explained in detail. Next, the common photocatalyst nanomaterials including titanium dioxide (TiO_2_), zinc oxide (ZnO), graphitic carbon nitride (g-C_3_N_4_), and graphene, were introduced based on their distinct structures, advantages, and limitations in water treatment applications. The synthesizing method of TiO_2_-based photocatalyst includes hydrothermal synthesis, sol-gel synthesis, and chemical vapor deposition synthesis. Different synthesizing methods contribute toward different TiO_2_ forms in terms of structural phases and surface morphology. The fabrication and performance of cellulose composite catalysts give readers a better understanding of the incorporation of cellulose in the development of sustainable and robust photocatalysts. The modifications including metal doping, non-metal doping, and metal–organic frameworks (MOFs) showed improvements on the degradation performance of cellulose composite catalysts. The information and evidence on the fabrication techniques of biomass-derived cellulose hybrid photocatalyst and its recent application in the field of water treatment were reviewed thoroughly in this review paper.

## 1. Introduction

Nowadays, the rising population on Earth has significantly amplified the production of organic solid waste [1]. The disposal of organic solid waste, such as biomass, creates a prolonged problem, especially in the agricultural industry [2]. Dumping solid waste into natural territory, such as landfills, has heightened the global waste level and rather induced certain risks regarding the handling method of solid waste [3]. At landfills, the degradation of biomass produces methane gases, and they are being released into the surroundings and further become a factor in the Greenhouse Gas (GHG) effect [4]. Such environmental issues have directed researchers towards discovering environmentally friendly and low cost methods to produce and commercialize potential materials from organic solid waste such as biomass [5]. In fact, organic solid waste such as biomass is also intrinsic to a high carbon value, so these types of waste should be redefined as “resources” [4]. Biomass has been regarded as a sustainable resource that could potentially minimize GHG emissions [5]. To acquire zero GHG emissions, there must be a balance between both the production of plant biomass and the management of its residual wastes. This can be acquired by establishing a circular economy which involves the activities of reuse, recycle, repurpose, and up-cycle [4,5].

Cellulose is a biopolymer that is widely available, renewable, and makes up the majority of biomass. Biomass materials including corn cobs, banana stems, sugarcane bagasse, and wheat straw are sources of cellulose. Depending on the source, lignocellulosic biomass contains 40–60% *w*/*w* cellulose, 15–30% *w*/*w* hemicellulose, and 10–25% *w*/*w* lignin [6]. The cellulose isolated from biomass is presented in the form of cellulose fibers. It has been claimed that the promising composition of agricultural residues, which contains more hemicellulose and less lignin than wood, promotes more effective nanofibrillation [7].

Among advanced oxidation processes, photocatalysis is commonly applied for the degradation of organic dye. It comprises a photo-oxidation reaction in the presence of photocatalysts under light irradiation. In photocatalysis, the photocatalyst is activated through the absorption of photon energy to accelerate the chemical reaction without being consumed [8]. When the energy received from light irradiation is equal to or exceeds the band gap energy of photocatalysts, the electrons in the valence band of photocatalysts will migrate to the conduction band of photocatalysts, leaving holes in the valence band of photocatalysts [9]. Simultaneously, the generated electrons and holes will carry out the reduction and oxidation reactions to produce superoxide anion radical and hydroxyl radicals, respectively. These reactive oxygen radicals may contribute to the oxidative pathways to degrade the organic pollutant molecules.

Despite the efficient removal of organic pollutants, conventional photocatalysis possesses limitations such as difficult recovery of the catalyst, generation of secondary pollution, and a high consumption of both catalysts and energy. To overcome these limitations, it is highly desirable to develop semiconductor photocatalysts with promising charge migration, high quantum efficiency, broad light spectral response, and good stability. Through the emergence of cellulose and its potential derivatives, the performance and sustainability of cellulose-based photocatalysts have advanced progressively in the past decade. Therefore, this review is able to provide insights and guidance for scientists who are looking for the development and functionality of cellulose-based nanostructured photocatalyst hybrids to address emerging environmental concerns.

## 2. Biomass

In general, biomass is known as a biological material obtained from plant-based or animal-based resources and their respective derived residues and wastes [5]. Biomass can be categorized into agricultural wastes, forestry wastes, industrial wastes, and municipal wastes [1,5,10,11]. Researchers tend to make use of biomass in researches via a facile route due to its recyclability and sustainability [3]. Both material costs and solid waste management risks can be lowered due to the utilization of renewable biomass as starting materials [12].

The global primary production of agricultural biomass is approximately 220 billion tonnes annually based on the dry weight basis [13]. Agricultural biomass mainly comprises cellulose, hemicellulose, and lignin as shown in Figure 1 [14]. Cellulose is a straight chain polymer comprising glucose monomers [15]. Hemicellulose has different short-chain polymers such as d-xylose, d-galactose, and d-glucose [16]. Lignin is a complex non-crystalline phenolic macromolecule with an amorphous nature and aromatic structures such as sinapyl, coniferyl, and coumaryl alcohols [15,16]. Furthermore, inorganic constituents such as calcium, potassium, silicon, magnesium, sodium, phosphorous, and chlorine can also be found in biomass [10,15,17]. Biomass contains about 51 wt% of carbon, 42 wt% of oxygen, and other remaining elements [10]. Biomass collected from agricultural waste is mostly lignocellulosic biomass and it can be utilized to synthesize carbon-based catalysts such as biochar and AC owing to high carbon content [2].

Looking into the marine biomass, it was reported that 17.4 million tonnes of mollusks and 8.4 million tonnes of crustaceans were produced globally in 2017 [18]. The examples of mollusks are mussel, clam, and oyster whereas the examples of crustaceans are lobster, shrimp, and crab. However, seafood wastes are often thrown back into the sea, burned, landfilled, or left aside to decompose [19]. The utilization of marine biomass has become an environmental priority and has amplified research regarding waste conversion into valuable products such as cellulose, chitosan, and chitin [19,20,21].

### 2.1. Biochar

Biochar is known as biomass-derived char and it is a carbon-rich material because it comprises about 60–90% carbon [22,23]. Biomass, such as agricultural residues (rice hulls and bagasse), garden residues, and municipal wastes, can be used to produce biochar [24]. The promising properties of biochar are the large specific surface area, high pore volume, high fertility, long-term stability structure, strong adsorption capacity, and enriched surface functional groups and mineral components [22,24,25].

Biochar can be produced via pyrolysis, gasification, and carbonization [26,27]. Pyrolysis is the conventional method to produce biochar and it is conducted within an oxygen-limited or oxygen-free environment at a temperature range of 300–800 °C [22,24,27]. During pyrolysis, the biomass components will experience both thermal reactions and molecular arrangements in order to construct a polymerized aromatic structure and this structure enables active compounds to functionalize biochar [23]. Meanwhile, gasification transforms biomass into a major gaseous product of syngas and a minor solid product of biochar at a temperature higher than 700 °C in the presence of an oxidizing agent [22,27]. Gases such as oxygen, air or steam can be applied as oxidizing agents [27].

Carbonization can be carried out prior to activation where biomass performs a thermal treatment (slow pyrolysis) to enrich the carbon content in the biomass [28,29]. The resultant solid residue (biochar) possesses low porosity and the formed pores are entrapped with tar-like materials [22,29]. The biochar can be further developed in activation processes even though the initial porosity of char is relatively low [28]. Carbonization parameters including carbonization temperature, heating rate, and residence time, contribute to the formation of the initial pore structure through the emissions of volatile matter from the carbon matrix [28,29].

Biochar is introduced into soil to improve soil fertility, enhance carbon sequestration, and mitigate greenhouse gas emission [22,24]. Other than that, biochar is also used as a precursor of synthesizing catalysts and contaminant adsorbents in both soil amendment and wastewater treatment [27]. The raw materials of biochar synthesis are cheap and abundant, while the preparation of biochar is cost-effective with lower energy requirements [24,25]. Thus, biochar is reported to be a potential low-cost and effective adsorbent in applications such as catalysis and soil remediation [25].

### 2.2. Activated Carbon (AC)

AC is defined as a form of amorphous carbonaceous materials with high surface area, high porosity nature, high adsorption capacity, and oxygenated functional groups [23,29,30,31]. The preparation of AC is close to the preparation of biochar except for the additional activation process and higher treatment temperatures are required [25,32]. The activation process can be carried out with three methods which are physical activation, chemical activation, and physiochemical activation [28,29,30]. The purpose of the activation process is to enlarge pore size, improve the pore volume, expand the pore diameter and increase the porosity [23,28]. Physical activation involves thermally treating the carbonaceous precursor in the presence of oxidizing agents such as inert gas, carbon dioxide, and steam, at elevated temperatures ranging from 400 to 1000 °C [29,30]. The introduction of an oxidizing agent promotes the internal porosity of AC [29].

For chemical activation, chemical agents will be added to the material prior to the same thermal treatment as physical activation [30]. The type and concentration of chemical agents, or named as activating agents, such as zinc chloride (ZnCl_2_), potassium hydroxide (KOH), phosphoric acid (H_3_PO_4_), and sodium carbonate (Na_2_CO_3_), are taken into account for the surface properties of AC. Nayak et al. [33] reported that the micropore volume and surface area of ZnCl_2_-prepared AC (0.61 cc/g and 2430.8 m^2^/g) was higher than KOH-prepared AC (0.32 cc/g and 1506.2 m^2^/g). Under similar activation conditions (precursor to activating agent mass ratio =1:0.5, 600 °C, and 1 h), ZnCl_2_ promoted the microporosity on AC, whereas KOH induced the development of larger pores on AC. Shrestha and Rajbhandari [34] found out that the AC impregnated with H_3_PO_4_ exhibited the highest surface area (1269.5 m^2^/g) as compared to AC impregnated with KOH (280.6 m^2^/g) and Na_2_CO_3_ (58.9 m^2^/g) at similar activation conditions (precursor to activating agent mass ratio =1:1, 400 °C, and 3 h) due to numerous mesopores and micropores. The activating agents such as KOH and Na_2_CO_3_ at the activation temperature of 400 °C were insufficient to induce a high porous structure on AC. The favorable activation temperature for acidic and alkaline activating agents were reported at 400–500 °C and 750–850 °C, respectively [35]. In addition, Zhang et al. [36] reported that the increasing mass ratio of sodium hydroxide (NaOH) to carbonized wheat bran from 1 to 5 at the activation temperature of 800 °C increased the mesoporosity of AC. Excessive usage of NaOH could damage the pore walls, widen the pore through violent etching, and destruct the micropore structure of AC.

Next, AC is usually utilized in applications of pollution control such as gaseous filter systems and wastewater treatment [23,30]. The high adsorption capacity of AC allows it to act as catalyst support, and to remove pollutants such as heavy metal and organic dyes from contaminated wastewater or dyeing unit effluent effectively [23,32,37]. Nevertheless, the drawbacks of commercial AC are high production cost, separation difficulty, low regeneration rate, and difficulty in reactivation process [31,32]. Similar to biochar, any low cost carbonaceous material can be used as promising precursor sources of AC and it can be obtained from woody biomass, agricultural waste, and forestry residues [29,30]. For example, coconut shells, bamboo, wood, silk cotton hull, coal, fruit peel, and especially biochar are used in the fabrication of AC [23,29,30,32,37].

Besides, Wickramaarachchi et al. [38,39] reported the fabrication of AC with hierarchical porous structure by using biomass such as mango seed husk and grape marc to develop sustainable supercapacitor materials. Both mango-seed-husk- and grape-marc-derived AC achieved high specific capacitance of 135 F/g and 139 F/g, respectively, at optimum conditions. Arun et al. [40] discovered that adding 0.1 wt% orange-peel-derived AC to the negative electrode of a lead acid battery cell increased capacity and raised charge acceptance along with lowering gassing voltages. When jute-fiber-derived AC was used as an anode material for lithium-ion batteries, it demonstrated a high specific capacity of 742.7 mA h/g after 100 cycles at 0.2 C [41]. This showed that the jute-fiber-derived AC was a stable and efficient anode material. Hence, biomass-derived AC can also be a potential material for energy storage applications.

### 2.3. Cellulose

Cellulose is defined as a linear chain of ringed glucose molecules forming a flat ribbon-like structure [42,43]. The molecular mass of cellulose ranges from 1.44 × 10^6^ to 1.8 × 10^6^ g and it shows thermal softening at a temperature of 231–253 °C [44]. Cellulose is a biopolymer made up of glucose monomers with the degree of polymerization ranging between 1 × 10^4^ and 2 × 10^4^ based on the source of cellulosic material [45,46]. Each glucose monomer contains three hydroxyl groups which control the crystalline packing and physical properties of cellulose [47]. The glucose linkages can be stabilized to form a linear cellulose chain because of the existence of hydrogen bonding between hydroxyl groups and oxygen atoms of the adjoining ring molecules [42]. These hydroxyl groups also define the chemical reactivity of cellulose, whereby their reactive sites enable the functionalization of cellulose materials [48]. Thus, chemical modifications of cellulose can enhance the adsorption property of cellulose towards pollutant removal by altering its physical and chemical properties [32].

Cellulose has degree of polymerization >2000, is insoluble in common solvents such as water, and is also weakly accessible to acid and enzymatic hydrolysis [32,49,50]. The material properties of cellulose are contributed by the phenomenon of aggregation via Van der Waals forces, and both inter- and intra-molecular hydrogen bonding inside the cellulose chains [46,51]. With the formation of inter-molecular hydrogen bonds, the cellulose chains are organized in a parallel arrangement to form stiff ribbon- or sheet-like structures through the hindering of free rotation of the rings on the glucose linkages [32,49,51]. The Van der Waals forces and weak CH–O bonds or intra-molecular hydrogen bonds hold these cellulosic sheets together in layers [44,52].

The linear cellulose chains, or named as cellulose fibrils, aggregate to form a microfibril, and subsequently the microfibrils are further assembled into the common cellulose fibers during the cellulose biosynthesis [46,51]. The range of microfibrils cross dimensions is between 2 and 20 nm, and this depends on the origin of cellulose [46]. Looking into the cellulose fibrils, cellulose chains are divided into crystalline and amorphous regions [43]. The cellulose chains in the crystalline region are arranged highly ordered, while the cellulose chains in the amorphous region are arranged disorderly. The crystalline region consists of a complex network of hydrogen bonds, which gives strength and toughness to cellulose fibrils [51]. The amorphous region, comprising cellulose chains with lower density compared to the crystalline region, has higher availability to form more hydrogen bonds with other molecules [53]. Eventually, the amorphous region can be easily hydrolyzed under harsh pre-treatment conditions to deliver nanosized cellulose [51,54].

Within the crystalline region, the variations of the inter- and intra-molecular hydrogen bonding and molecular orientations generate cellulose polymorphs or allomorphs [55]. The generated cellulose polymorphs are influenced by the source, extraction method, and treatment of cellulose [46]. Native cellulose is known as the primary cellulose produced from natural biosynthesis, which does not undergo any changes of form and it has the polymorph of cellulose I [47,56]. Cellulose I can be transformed into cellulose II by either mercerization in strong alkali medium or acid regeneration [56]. Cellulose I and cellulose II can also be converted into the respective cellulose III_I_ and cellulose III_II_ through ammoniacal treatments [57].

Cellulose I is commonly studied due to its high abundance in nature [56]. Cellulose I comprises two crystalline forms or suballomorphs which are Iα and Iβ lattices [46,49]. The Iα lattice has a one-chain triclinic structure while the Iβ lattice has a two-chain monoclinic structure [45,52]. Next, different cellulose sources vary in the fractions of Iα and Iβ crystal structures [45]. The Iα crystal structures predominate in bacterial cellulose and algae cellulose, whereas the Iβ crystal structures predominate in plant cellulose and tunicate cellulose [58].

It is interesting that both Iα and Iβ crystal structures comprise stacks of sheet-like cellulose chain layers to form three-dimensional (3D) crystals [52,56]. The Iα and Iβ crystal structures have comparable unit chain length, interchain distance, and intersheet distance, which are 10.4 Å, 8.2 Å, and 3.9 Å, respectively [52]. Nevertheless, the Iα and Iβ crystal structures differ from the framework of relative displacement between adjacent cellulosic layers [56]. The Iα crystal structure can be converted into Iβ crystal structure through annealing and it is an irreversible process [45,49]. Thus, this shows that the Iβ crystal structure is more stable compared to the meta-stable Iα crystal structure [49,56].

### 2.4. Chitosan and Chitin

Chitosan is known as poly-β-(1,4)-2-amino-2-deoxy-glucopyranose whereby it is a polysaccharide consisting of glucosamine and N-acetylglucosamine as copolymers [59,60]. Chitosan is produced from the partial deacetylation of chitin [19,59]. Typically, chitosan is determined as the second abundant biomass-derived polysaccharide which is low cost, environmentally friendly, biocompatible, and biodegradable [60]. Chitosan has two monosaccharide units where their proportions will be affected by the subsequent alkaline treatment [20,21]. Chitosan consists of primary amine and free hydroxyl groups [21]. Besides the acetylamine or free amino groups which replace the hydroxyl group at the C-2 position of the cellulose structure, chitosan has a similar structure compared to cellulose as shown in Figure 2 [61,62,63]. The presence of highly reactive amino groups promotes the generation of intra- and inter-molecular hydrogen bonds with the abundant hydroxyl groups which form linear aggregates and rigid crystalline domains [20,21]. This resulted in the high viscosity and exhibition of polymorphism of chitosan [21,61].

Crustacean shells are the primary chitin source for the industrial production of chitosan [19]. Generally, chitin is a structural element that can be found in crustaceans, exoskeletons of insects, and cell walls of fungi as shown in Figure 3 [60]. Although marine animals are still the main source of chitin, the fungal chitin obtained from mushrooms is gradually increasing nowadays. It is the second most abundant naturally occurring biopolymer after cellulose [65]. The chitin within these seafood wastes has a slow rate of biodegradation and this causes the yielding of large piles of processing discards from seafood processing plants [21].

Chitin can be deacetylated into chitosan by chemical and enzymatic methods [67]. The chemical method includes demineralization, deproteinization, deacetylation, and decolorization [19]. The demineralization step removes calcium carbonate (CaCO_3_) from the crustacean shells with dilute hydrochloric acid (HCl), whereas deproteinization solubilizes the protein with dilute aqueous NaOH [21]. Then, the produced chitin can be treated with hot concentrated NaOH for a long period to synthesize chitosan through heterogeneous and homogeneous processes with 85–99% and 48–55% degree of deacetylation, respectively [62,67,68]. The alkaline treatment hydrolyzes the acetyl groups and converts N-acetylglucosamine into glucosamine with free amino groups. The degree of deacetylation indicates the glucosamine to N-acetylglucosamine ratio whereby chitin transforms into chitosan. When the proportion of glucosamine is higher than N-acetylglucosamine, the produced compound is called chitosan, and vice versa for chitin [19]. The additional decolorization step can be carried out to remove color and improve physical appearance [21].

Furthermore, the enzymatic method for producing chitosan uses enzymes under mild conditions [62]. Enzymatic methods utilize enzymes extracted from *Mucor roxii* and *Absidia coerulea* for the deacetylation of chitin [67]. Chitinases or chitin deacetylases can be obtained from *Mucor roxii* and *Absidia coerulea*, and these enzymes have a good thermostability which could act optimally at 50 °C [69]. The utilization of chitin deacetylases is mainly to produce novel and well-defined chitosan oligomers [68]. In comparison with the enzymatic method, chemical methods have shorter processing time and higher suitability for mass production [67,68,70]. Therefore, the chemical method is usually preferred to produce chitosan.

Besides viscosity, both the degree of deacetylation and molecular weight of chitosan affect the solubility, reactivity of proteinaceous material coagulation, heavy metal ion chelation, and physical properties of chitosan films [62]. Chitosan is considered a multifunctional polymer that can be used in food preservatives, tissue engineering, biocatalysis, and anticancer applications [21,68,71]. The main drawback of chitosan is poor solubility at physiological pH value 7.4 owing to the partial protonation of the amino groups [21,62]. Hence, chitosan can be further modified with processes such as grafting, cross-linking, composites, and substituent incorporation [21]. In energy storage applications, Ramkumar and Minakshi [72] reported the fabrication of cobalt molybdate modified by using chitosan cross-linked with glutaraldehyde as a cathode material in a hybrid capacitor. Cross-linking has the main benefit of enhancing the surface functionality of the modified electrode [73]. The degree of amorphosity in the composite was impacted by the cross-linking of chitosan and glutaraldehyde, which reduced particle size and enhanced the development of cluster-like particles that produced a capacitance that was about four times greater than as-prepared cobalt molybdate. The modified electrode also exhibited outstanding cycling stability with 97% coulombic efficiency (over 2000 cycles), indicating that chitosan gel adheres firmly to the molybdate moiety of cobalt molybdate.

## 3. Types of Cellulose

Cellulose is classified as the most abundant and renewable biopolymer found in nature [49]. It is present in biomass such as plants, bacteria, and animals (tunicates) [42,44]. Cellulose produced from biomass in nature is around 1 trillion tonnes annually, and this proves that it is an inexhaustible source of raw material [74]. Regardless of the sources, cellulose is a biopolymer composed of β-d-glucopyranose (glucose) monomers held by linear β (1–4) linkages and the repeating unit is named as cellobiose in which a dimer of glucose [46,75]. The term β is determined from the position of the ether oxygen which is located on the same side of the glucose rings with the hydroxyl groups [76]. In the glucose rings, the hydroxyl groups are arranged in equatorial positions, and they possess important features such as controlling the crystalline packing, stabilizing the glucose linkages, and determining the chemical reactivity of cellulose [44,76].

Other than abundant and renewable properties, cellulose is extensively used in its natural purified state or derivatives because it is cheap, environmentally friendly, biocompatible, and readily available [42,49,50]. Cellulose fibers, microcrystalline cellulose (MCC), cellulose nanofibers (CNF), nanocrystalline cellulose (NCC), cellulose hydrogels and aerogels, and cellulosic composites are the examples of cellulosic derivatives [43,51]. High surface area, low density, high aspect ratio, good mechanical properties, low cost, and adaptable surface properties contribute towards the utilization of cellulose in composites, polymers, synthetic fibers, and antibodies [42,77]. Cellulose and its derivatives are used as contaminant adsorbents and stabilizers for active particles in water treatments to remove organic and inorganic pollutants [78]. Their global market value is predicted to hit $1.08 billion by 2020, which benefits the pharmaceutical and food divisions [48]. Hence, the cellulose in biomass is a suitable carbon source to replace commercial synthetic applications.

### 3.1. Plant Cellulose

Plants are the primary reserves of cellulose [55,74]. Plants are categorized into wood (e.g., hardwood, softwood, cotton linter) and non-wood types (e.g., agricultural biomass) [48]. Agricultural biomass is in high abundance and readily available. Based on the structure of agricultural biomass, lignin is located in the outer plant cell wall and cellulose is located within the lignin shell along with hemicellulose [79]. Hemicellulose and lignin are mostly bonded with cellulose via hydrogen bonds or covalent bonds [79,80]. Lignin is has a resistance towards biological attack and stiffens the plant stem to protect it from external forces (e.g., wind), whereas hemicellulose supports the compatibility between lignin and cellulose [81].

In general, the plant cell wall is divided into primary and secondary walls. The secondary wall is responsible for the overall characteristics of plant fibers as cellulose is mainly located in the secondary wall [75]. The secondary wall consists of three layers where the middle layer mainly contributes to the mechanical properties of cellulose fibers [81]. Inside the middle layer of secondary plant cell wall, the cellulose fibrils impart rigidity and maximum tensile and flexural strengths upon their alignments along the length of plant fibers [47]. There are usually 30–100 cellulose molecules aligned helically in the extended cellulose chain configuration [81]. In short, cellulose helps to maintain the plant cell wall structure with its appealing mechanical properties [46].

Cellulose fiber exists in terms of cellulose chain groups inside the lignin matrix as it does not occur naturally as an isolated molecule [47,75]. The synthesis or isolation of cellulose particles require purification and mechanical treatments in order to remove matrix materials (e.g., hemicellulose and lignin) partially or completely, isolate the cellulose fibers, and promote uniform reactions in the subsequent treatments [43]. Agricultural biomass, or in other words, lignocellulosic biomass, undergoes chemical pretreatments (e.g., acid hydrolysis, alkali treatment, acid-chlorite treatment, organosolv treatment), biological pretreatments, and mechanical pretreatments (e.g., homogenization, grinding processes) [82,83]. The final cellulose product is greatly influenced by the concentration of chemicals, reaction time, and temperature.

Upon chemical pretreatment, acid hydrolysis is widely applied in the production of MCC and NCC [83,84]. Table 1 shows the preparation of MCC and NCC from various sources using different acid hydrolysis methods. Acid hydrolysis uses mineral acids such as sulfuric acid (H_2_SO_4_), H_3_PO_4_, and HCl [46]. For example, weak H_2_SO_4_ (below 4 wt%) hydrolyzes the polysaccharide hemicellulose completely into monosaccharide xylose by breaking the xylosidic bonds [54]. Strong H_2_SO_4_ hydrolyzes the amorphous regions of cellulose fibrils through the esterification of hydroxyl groups by sulphate ions, and yields highly crystalline nanosized cellulose particles in the remaining treated solution [82,83].

For acid hydrolysis, prolonged time is necessary to achieve a complete reaction [95]. However, a prolonged reaction time can hydrolyze hemicellulose and some extent of cellulosic materials as cellulose degrades into water-soluble glucose molecules [54,95]. It was reported that microwave-assisted acid pretreatment minimized the reaction time remarkably [54]. The subsequent washing by water or NaOH was usually used to neutralize the pH for the treated cellulose [83]. The main drawback of acid hydrolysis is the generation of acid-containing effluent, which requires additional treatment before disposal in the environment [95]. This is because acid is corrosive and toxic, and it is extremely harmful to the environment. Hence, the overall effect of acid hydrolysis towards lignocellulosic biomass is influenced by acid–biomass ratio, acid concentration, and reaction temperature and time [54].

Alkali treatment and acid-chlorite treatment are mainly employed to remove lignin from lignocellulosic biomass [83]. Alkali treatment uses mediums such as NaOH, potassium hydroxide (KOH), calcium hydroxide (Ca(OH)_2_), sodium carbonate (Na_2_CO_3_), and ammonia [54]. Lignin and silica can be dissolved in alkali treatment by breaking down uronic and acetic esters linkages, which causes cellulose swelling [96]. In addition, alkali treatment also hydrolyzes hemicellulose partially to reduce the crystallinity of cellulose, and increases the internal area and porosity of cellulose [54]. Better delignification effect is also obtained from alkali treatment as compared to acid hydrolysis [54]. The efficiency of the alkaline treatment can be improved by using the combination of both alkali and acidic treatments with low amounts, and it is more economical and environmentally friendly [84].

Besides, acid-chlorite treatment involves the combination of sodium chlorite (NaClO_2_) and glacial acetic acid (CH_3_COOH) [83]. The acidified NaClO_2_ delignifies and bleaches the plant fibers until the product becomes white and free of lignin. Normally, the bleaching process is employed when incomplete delignification of the lignocellulosic materials occurs, which yields brown-colored cellulosic products [54]. The purpose of bleaching is to eliminate the remaining lignin and hemicellulose contents embedded in the obtained cellulosic products. The commonly used bleaching agents are chlorine and hypochlorite compounds (e.g., sodium hypochlorite (NaClO), NaClO_2_) due to their economical production of high bright chemical pulps [97]. Modifications to the bleaching stage are made to deal with the generation of effluents containing carcinogenic and mutagenic chlorinated compounds after the bleaching process [54]. Table 2 shows different combined treatment methods to extract cellulose from various lignocellulosic biomass sources.

Another alternative delignification method for lignocellulosic biomass is organosolv treatment or organosolv pulping process. Organosolv treatment uses organic solvents such as ethanol, CH_3_COOH, and acetone [96]. For instance, OPEFB was treated with a mixture of aqueous ethanol and diluted H_2_SO_4_ at 120 °C for 1 h, and followed by the treatment of diluted hydrogen peroxide (H_2_O_2_) at 50 °C for 4 h to remove lignin and hemicellulose [103]. The distillation of organic solvent could isolate the lignin content from the lignocellulosic materials by using a mixture of organic solvent and water [96].

Biological pretreatment is employed to produce cellulose with high purity and crystallinity [54]. Typically, the removal of protective layers (lignin and hemicellulose) is required to allow easier access to cellulose. Microorganisms (e.g., white rot fungi, soft rot fungi) attack both lignin and hemicellulose, while cellulase-less mutant is generated for the selective degradation of lignin over cellulose [96]. Even though biological pretreatment is energy-saving and requires no chemicals, it still suffers the main drawback such as low degradation rate and efficiency for lignin and hemicellulose [54,96].

Mechanical pretreatment is applied in the production of microfibrillated cellulose and nanofibrilated cellulose [82]. The obtained cellulose fibers break into smaller sizes through high pressure homogenizing or grinding processes. For instance, the cellulose fibers are consistently agitated with a high shear homogenizer for some time which results in nanofibrils [81]. Nevertheless, mechanical treatment is highly energy-intensive which leads to alternative methods such as enzymatic and acetylation treatments [82].

Lately, agricultural biomass is the most cited and it is a preferred substrate to isolate cellulose and its derivatives. The broad range of cellulose-based raw materials includes corn straw, OPEFB, rice husk, durian shell, wood, cotton, potato tubers, and soybean stock [43,51]. The average cellulose contents in wood, cotton, jute, flax, and ramie, are 40–50 wt%, 87–90 wt%, 60–65 wt%, 70–80 wt%, and 70–75 wt%, respectively [44]. Plant cellulose is mainly utilized in the textile, pulp and paper, packaging, and pharmaceutical industries for harvesting, processing, and handling [43]. Nevertheless, plant cellulose has disadvantages, such as being easily degraded at temperatures above 200 °C, and a high moisture content, which hinders the subsequent carbonization heat treatment [47]. Overall, the benefits of plant cellulose surpass its shortcomings and it is commonly chosen for mass production compared to bacterial cellulose due to lower costs [82].

### 3.2. Bacterial Cellulose

Bacterial cellulose is firstly reported as *Bacterium xylinum* by Brown in 1886 [104]. The bacterial cellulose is initially produced by bacterial strains named *Gluconacetobacter xylinus* (*G. xylinum*), previously known as *Acetobacter xylinum* (*A. xylinum*) [76]. Researcher Brown collected *G. xylinum* from a pellicle formed on the surface of beer [105]. *G. xylinum* can also be found in the fermentation of carbohydrates like rotten fruits and unpasteurized wine [106]. Other bacteria species such as *Agrobacterium*, *Pseudomonas*, *Rhizobium*, and *Sarcina*, are also able to produce bacterial cellulose [107]. Different bacteria species synthesize cellulose with various morphologies, structures, properties, and applications [108]. In comparison with other bacteria strains, *G. xylinum* is commonly used to produce bacterial cellulose commercially due to its relatively high productivity [109]. Regarding productivity, one bacterium is predicted to transform 108 glucose molecules per hour into cellulose [76].

Furthermore, *G. xylinum* is the most studied source and it will be used to demonstrate the proposed biochemical pathway from glucose to cellulose [107,110]. Cellulose is synthesized extracellularly, which is between the outer and cytoplasma membranes of *G. xylinum* [111]. The biosynthesis of bacterial cellulose requires cellulose synthase. The cellulose synthase is activated by the enzyme named cyclic diguanylmonophosphate (c-di-GMP) [107]. An individual *G. xylinum* cell could polymerize up to 200,000 glucose molecules per second into cellulose chains [112].

Looking into the formation of bacterial cellulose, an elongated cellulose chain is formed through the aggregation of about 6–8 glucose chains [111]. It is reported that there are approximately 50 to 80 pores arranged along the long axis of the *G. xylinum* cell [112]. These elongated cellulose chains or fibrils escape from the pores (diameter about 3 nm) on the surface of *G. xylinum* cell and are further assembled into ribbon-like microfibrils [106,111]. The bundles of ribbon-like microfibrils aggregate into a network of interwoven ribbons and finally form a bacterial cellulose membrane or pellicle [111]. Moreover, *G. xylinum* produces the pellicle on the surface of the liquid culture medium. The formed pellicle is a thick gel comprising ribbons of cellulose microfibrils and 99% water [107]. *G. xylinum* is an aerobic bacteria and it produces pellicle to protect itself from ultraviolet light [111]. The pellicle acts as a barrier to safeguard against other organisms and heavy-metal ions, and yet allows the diffusion of nutrients into the bacteria cell [105]. Therefore, the biosynthesis mechanism helps *G. xylinum* to reach the oxygen-rich surface for survival purposes.

The synthesis of bacterial cellulose focuses mainly on culturing methods and purification to promote cellulose microfibrillar growth and to eliminate the bacteria and other media, respectively [43]. Purification of bacterial cellulose normally serves the purpose of killing the bacteria and removing unwanted byproducts by using standard NaOH treatment [76]. The culture medium and culture conditions are the crucial factors in each type of the culturing method. In the culture medium, bacteria strain, carbon source, nitrogen source, nutrition, pH, and oxygen delivery affect the bacteria growth and properties of bacterial cellulose [108,113]. The optimum pH value of the culture medium ranges from 4 to 7 [76]. It is reported that bacteria perform efficiently in culture media containing rich carbon source and limited nitrogen source [113]. Meanwhile, the bacteria synthesize cellulose ribbons containing mixtures of Iα and Iβ lattices under different culture conditions including stirring, temperature, and additives. The Iα/Iβ ratio and width of cellulose microfibrils can be changed without the presence of additives [76].

The common methods to prepare bacterial cellulose are static, agitated/shaking, and bioreactor cultures [108]. For static culture method, bacterial cellulose is aerobically synthesized in a thick cellulosic surface mat (pellicle) on the surface of the culture medium [112,114]. The overall yield of bacterial cellulose pellicle or hydrogel sheets depends on the surface area of the gas–liquid interface [115]. The formation and growth curve of bacterial cellulose is identical despite different culture media. The bacterial cellulose microfibers accumulate to form a hydrogel sheet structure. The bacteria strains help to form and grow bacterial cellulose hydrogel sheets along with the consumption of nutrients. Glucose and CH_3_COOH are the nutrients required for the bacteria strains [108]. When the bacterial cellulose gel sheet thickens, the oxygen supply is limited to the bacteria strains which inhibit the further synthesis of bacterial cellulose. As a result, the produced bacterial cellulose has high porosity. A long culture time and intense labor force cause low productivity of bacterial cellulose [116]. Static culture is the standard method to synthesize bacterial cellulose at lab-scale due to its simplicity and requires low shear force [108].

Agitated/shaking culture is introduced to promote bacteria cell growth [76]. Agitated/shaking culture provides shear force to the culture medium during the synthesis of bacterial cellulose. Upon shaking or agitation, forced aeration improves the respiration of bacteria to enhance the synthesis of bacterial cellulose [116]. Instead of forming hydrogel sheet structures, small irregular shaped (sphere-like) bacterial cellulose pellets are usually formed throughout the medium by continuously mixing with oxygen in the shaking culture method [117]. The size, geometry, and internal structure of the bacterial cellulose pellets are affected by the shaking or rotational speed. It is reported that hollow sphere-like bacterial cellulose pellets were obtained at 150 rpm whereas solid sphere-like bacterial cellulose pellets were obtained at 125 rpm [118]. The bacterial cellulose produced from agitated/shaking culture method has a lower degree of polymerization, crystallinity, and mechanical properties as compared to static culture [115]. In spite of increased oxygen delivery, agitated/shaking culture has lower productivity of bacterial cellulose than static culture due to the generation and accumulation of cellulose-negative mutants [116]. Only bacterial species (e.g., *G. xylinum*) that are resistant towards the mutant generation effect could result in higher productivity of bacterial cellulose [117].

Basic static and agitated/shaking cultures could not provide uniform mixing and oxygen delivery in the culture medium [117]. Their batch mode processes prevented the addition of supplementary nutrients which affects the mass production of bacterial cellulose. The bioreactor culture method uses specially designed reactors along with high-speed agitation. Small-sized bacterial cellulose granules are formed by rapid rotation speed and moving shafts within the bioreactors [117]. The examples of bioreactor are stirred tank bioreactor, airlift bioreactor, and rotating disc bioreactor. High energy consumption of stirred-tank bioreactors produce fibrous bacterial cellulose suspensions with a low degree of polymerization, low crystallinity, and low elastic modulus in comparison with bacterial cellulose hydrogel sheets, owing to high agitation and control of oxygen transfer [119]. The airlift bioreactor provides adequate oxygen supply to the culture medium and it requires significantly lower energy consumption than the stirred-tank bioreactor [120]. Rotating disc bioreactor produces a homogenous bacterial cellulose structure where the circular discs rotate to promote continuous interaction with air and liquid media [108]. The bacteria attach at the surface of the circular discs and absorb both the nutrients inside the culture medium and oxygen at the surface of the culture medium to synthesize bacterial cellulose [114].

The interesting properties of bacterial cellulose, such as outstanding mechanical properties, high moisture permeability, and biocompatibility, allows it to be used in different applications especially in the bioengineering field [121]. Bacterial cellulose fibers are used as reinforcing agents in composite due to their inert property [81]. The promising biocompatibility expanded the use of bacterial cellulose as a commercial wound care dressing material. Bacterial cellulose and its derivatives are also potential materials for scaffolds, drug-delivery systems, membrane, and filter materials [122]. Although bacterial cellulose has many advantages, it is still very expensive to produce in mass production [110]. The usage of various waste media and carbon sources produces bacterial cellulose with similar physicochemical properties as compared to commercial H–S media [108]. The promising waste materials allow the production of bacterial cellulose to be more economical and environmentally friendly. Hence, future research is set to develop commercial production of bacterial cellulose [122].

### 3.3. Comparison between Plant Cellulose and Bacterial Cellulose

Bacterial cellulose has an identical molecular structure to plant cellulose [119]. However, the chemical and physical properties of bacterial cellulose are different from plant cellulose. Bacterial cellulose has a higher purity which is pure or nearly 100% of cellulose content compared to plant cellulose (60–70%) [113]. The reason is that plant fibers normally contain polymers such as lignin, hemicellulose, and pectin, and also functional groups such as carbonyl and carboxyl, that are bonded with plant cellulose upon isolation and purification processes [106]. Other than the hydroxyl groups, bacterial cellulose is free of any other biopolymers or functional groups and eventually it does not require additional purification steps.

Cellulose fibrils in plant cellulose are formed within the plant cell wall matrix, whereas cellulose fibrils in bacterial cellulose are formed extracellularly and they are metabolically inert [112]. The extracellular synthesis and nanosized bacterial cellulose promote a stronger hydrogen bonding between cellulose fibrils than plant cellulose [81]. The degree of polymerization of bacterial cellulose and plant cellulose are 2000–6000 and 13,000–14,000, respectively [107]. The crystallinity of plant cellulose nanofibers ranges from 36 to 91% [123]. In comparison with plant cellulose, bacterial cellulose on average has a higher crystallinity of up to 90% with dominant Iα crystal structures [124]. The higher thermal stability of bacterial cellulose is attributed to its high purity and crystallinity [113]. This allows the treated bacterial cellulose to have a maximum decomposition temperature of 350–355 °C [76].

Furthermore, the hydroxyl groups located on the surface of plant cellulose and bacterial cellulose contribute to their hydrophilicity properties. The larger surface area of bacterial cellulose imparts a higher liquid loading capacity than plant cellulose. Bacterial cellulose acts as a hydrogel and has higher water absorbing and holding capacities because it contains high water content, which is 90% and above [77,110]. The porosity of plant cellulose is improved by chemical treatment. The dense 3D network of fibrils assembled in the structure of bacterial cellulose forms porous sheets [114]. The random orientations in the 3D network are caused by the biosynthesis of bacterial cellulose [76]. In addition, both tensile strength (20–300 MPa) and Young’s modulus of bacterial cellulose (sheet: 20,000 MPa; single fiber: 130,000 MPa) are more notable than that of plant cellulose [108]. With these properties, bacterial cellulose acts as a suitable thermal stabilizer in resins [113].

### 3.4. Other Cellulose Types

Cellulose can also be obtained from algae [125]. Algae are the fastest growing plant on Earth and they are branched into macroalgae (e.g., seaweeds) and microalgae [126]. With the unlimited and free sunlight from the sun, they grow rapidly by transforming solar energy into biomass effectively via photosynthesis. Macroalgae are also differentiated in colors based on their respective natural chlorophylls and pigments such as green seaweed (Chlorophyta), red seaweed (Rhodophyta), and brown seaweed (Phaeophyta) [127]. Cellulose is found in the cell walls of algae and it plays an important role as the building block of structural support in algae [128]. Other than cellulose, the cell walls of algae also comprise mannans, xylans, and sulfated glycans [126]. The functional groups of carboxyl, hydroxyl, amino, and sulfate can be found in these components [128]. They are responsible for the adsorption capability of algae in treating heavy metals [129].

The biosynthesis of cellulose mostly occurs at the plasma membrane of algae, except for those algae species that synthesize cellulose scales [130]. The linear and rosette-like terminal complexes are responsible for the biosynthesis, polymerization, and crystallization of cellulose [130]. Moreover, they play a role in assembling cellulose microfibrils [131]. For instance, *Valonia algae* contain large terminal complexes with nearly 10 catalytic sites to produce about 10 nm microfibrils [132].

The treatment methods for algal cellulose sources essentially incorporate culturing methods and purifications to eliminate the algae cell wall matrix [43]. Similar to plant cellulose and bacterial cellulose, NCC can be extracted from algae by utilizing acid hydrolysis, enzymatic hydrolysis, and mechanical treatments [133]. It was reported that the NCC produced from red algae waste through acid hydrolysis has high mechanical performance and good transparency [134]. Since enzymatic hydrolysis is more environmentally friendly and achieves a higher glucose yield than acid hydrolysis, it is more preferable to treat algal biomass for bioethanol production [126]. Algae and its derivatives have good biocompatibility and biodegradability. They are potential materials to construct hybrid and composite materials. In biomedical applications, algae-based polyesters are used as scaffolds and as controlled release of pharmaceutical agents [135].

Other than that, the animal source for cellulose is tunicate [136]. The name Tunicata is derived from a special integumentary tissue called a tunic, which encloses the whole epidermis of the animal cell [137]. The subphylum Tunicata is categorized into three classes which are *Ascidiacea*, *Thaliacea*, and *Appendicularia*. *Ascidiacea* (sea squirts) and *Thaliacea* contain tunics while *Appendicularia* does not contain tunic and yet produces cellulosic materials [136]. Tunic mainly consists of polysaccharides especially tunicin, in other words, tunicate cellulose, and proteins (e.g., collagen, pectin) [138]. The tunic is responsible for phagocytosis, pigmentation, colonial allorecognition, bioluminescence, photosymbiosis, innate immunity, chemical defense, tunic contraction, and impulse conduction in terms of biological functions [139].

Tunicate cellulose is chemically identical to both plant cellulose and bacterial cellulose. It is formed by the cellulose synthase which is found in the plasma membrane of the epidermal cells [136]. Besides cellulose biosynthesis, the cellulose synthase also contributes to the proper formation of tunic tissues and metamorphic events [140]. Similar to algae cellulose, cellulose is responsible for the skeletal structure in the tunic tissues. The hundreds of cellulose microfibril bundles are deposited in a multilayer pattern which is parallel to the epidermis surface [136]. The shape and dimensions of the cellulose microfibrils are influenced by their respective species. In the Ascidiacea class, mostly ascidians are reported to possess cellulose I microfibrils in the tunic tissues [137].

The synthesis of tunicate cellulose includes the isolation of mantel from the animal and the isolation of cellulose fibrils from the protein matrix [43]. In a simple procedure, the obtained tunicate tunic will be treated with acid hydrolysis, kraft cooking (involving alkaline treatment), and bleaching [136]. The purification of tunicate cellulose uses alkali solutions (e.g., NaOH and KOH) and acidic solutions (e.g., CH_3_COOH, nitric acid) at elevated temperatures [136]. The tunicate cellulose also can be treated with strong acid hydrolysis to prepare NCC. The collected tunicate NCCs had lengths of 500–3000 nm, widths of 10–30 nm, and aspect ratios of 10–200 [141]. Thus, tunicate cellulose is another promising material for the preparation of NCC and composite film applications [142].

Tunicate cellulose possesses a high specific surface area (150–170 m^2^/g), high crystallinity (95%), and a reactive surface containing hydroxyl groups which promote good mechanical properties [143]. Modified NCCs are widely used in the biomedical engineering field, such as in scaffolds and biomarkers. The tunicate NCCs with high aspect ratios are more difficult to detach from the cell surface compared to smaller cotton NCCs during the inhalation studies [144]. Furthermore, the tunicate NCC membranes fabricated by using the vacuum-assisted self-assembly method had high mechanical strength, distinguished pH- and temperature-stability, good cycling performance, and achieved highly efficient separation of oily water [145]. Therefore, tunicate species like *Ciona intestinalis* could be farmed at a large scale to produce commercial tunicate cellulose for the production of chemicals, materials, and biofuels [136].

## 4. Photocatalyst Nanomaterials

TiO_2_, ZnO, g-C_3_N_4_, and graphene are the common semiconductor photocatalysts applied in the degradation of organic pollutants. In general, a semiconductor photocatalyst consists of a band gap that separates the valence band and conduction band. The band gap energy determines the applicability of a semiconductor in photocatalysis [146]. The semiconductor photocatalysts with wide band gap energies depend on the electron excitation by obtaining additional energy from ultraviolet (UV) light radiation. The electrons excited from the valence band to the conduction band induce charge separation. Hence, the electron–hole pairs are formed to engage in the redox reactions for the degradation of organic pollutants [147]. Semiconductor photocatalyst often faces the significant drawback of the rapid recombination of the electron–hole pairs [148]. This hinders the generation of hydroxyl radicals (•OH) which play a crucial part in photocatalytic degradation.

### 4.1. Titanium Dioxide (TiO_2_)

The application of TiO_2_ electrode was first prepared in 1967 and it had demonstrated heterogeneous photocatalytic oxidation through the splitting of water molecules under UV light radiation [149]. To date, TiO_2_ is the most studied semiconductor photocatalyst for the removal of dyes and phenolic compounds from wastewater [150]. This is because of its low toxicity, low cost, high chemical stability, and high thermal stability [151]. The high stability of TiO_2_ allows it to stand out from other semiconductors such as gallium phosphide and cadmium sulfide, which generate toxic byproducts [152]. TiO_2_ is applied in commercial applications and products (e.g., cosmetics, catalysts, desiccant) due to inert and long-term photostability properties [153].

Besides that, TiO_2_ also possesses high UV absorption and superhydrophilicity which is important for the photocatalytic degradation of organic pollutants and solar fuel production [154]. TiO_2_ is thermodynamically efficient versus normal hydrogen electrode (NHE) at pH 7 [155]. The photogenerated electrons demonstrate higher reduction strength due to the more negative conduction band potential of TiO_2_ which is −0.5 V. Meanwhile, the generated holes demonstrate higher oxidation strength due to the more positive valence band potential of TiO_2_ which is + 2.7 V.

Despite these potentials, there are some limitations of TiO_2_ that affect the performance of TiO_2_ in the decomposition of organic chemical compounds. Firstly, the large band gap of TiO_2_ limits the application of TiO_2_ in certain photocatalytic degradation processes. It can only absorb about 4–5% of the solar spectrum. Secondly, the fast recombination of photogenerated electron–hole pairs lowers the photocatalytic efficiency of TiO_2_ [156]. Thirdly, the weak adsorption of organic pollutants on the surface of TiO_2_ resulted in the poor affinity of TiO_2_ and slower rate of photocatalytic degradation [150]. Fourthly, TiO_2_ nanoparticles aggregate easily, owing to the large surface area-to-volume and surface change in certain media [157]. The aggregation of TiO_2_ nanoparticles forms aggregates or clusters with sizes hundreds of times bigger than their initial sizes. These TiO_2_ aggregates obstruct active sites from exposure to light radiation and eventually inhibit photocatalytic activity [150]. Lastly, the recovery of TiO_2_ nanoparticles remains a revolving issue in wastewater remediation. To prevent the release of free nanoparticles into the water, TiO_2_ is synthesized in thin films or immobilized on substrates [147].

Next, the band gap of a photocatalyst greatly influences its catalytic performance. Figure 4 illustrates the band gap energy of different photocatalysts. The redox potential level of adsorbate species and band gap energy govern the possibility and rate of charge transfer [158]. The conduction band energy of TiO_2_ is located slightly higher than the reduction potential of oxygen molecules (O_2_), which makes it easier for the electron migration from the conduction band of TiO_2_ to O_2_ [147]. Meanwhile, the valence band energy of TiO_2_ is located lower than the oxidation potential of most electron donors. This allows the transfer of oxidative holes to the •OH radicals adsorbed on TiO_2_ surface and eventually enhances the redox reactions of pollutant degradation.

TiO_2_ is an n-type semiconductor, and it consists of three crystal structures which are anatase, brookite, and rutile as shown in Figure 5. In these crystal structures, the titanium atoms coordinate with six oxygen atoms to form TiO_6_ octahedron units [151]. The 3d orbitals of titanium atoms form the lower part of the conduction band of TiO_2_, while the overlapping of 2p orbitals of oxygen atoms forms the valence band of TiO_2_ [155]. The band gap energies of anatase, rutile, and brookite are determined around 3.2 eV, 3.0 eV, and 3.1 eV, respectively [160]. The electron excitation under UV light radiation at wavelengths below 400 nm is necessary due to the large band gap energies of TiO_2_ crystal structures [152]. These crystal structures or polymorphs influence the photocatalytic activity of TiO_2_. The photocatalytic activity of a semiconductor is mainly dependent on the light absorption, reduction and oxidation rates, and electron–hole recombination rate [161].

In the anatase crystal structure, the TiO_6_ octahedras are shared or connected through their corners (vertices) [151]. In other words, the octahedras share four edges which results in the tetragonal structure of anatase [162]. In anatase TiO_2_, the Ti−Ti lengths are longer and the Ti−O lengths are shorter compared to rutile TiO_2_ [161]. The anatase crystal structure is normally found in solution-phase preparation methods of TiO_2_ like sol-gel process [152,160]. Anatase is only stable at low temperatures. It is reported that anatase is thermodynamically stable in equivalent-sized TiO_2_ nanoparticles with sizes smaller than 11 nm [161]. Upon, calcination, anatase will transform to rutile at temperatures above 600 °C.

Next, the defects on the TiO_2_ affect the reduction activity. The defects tend to trap the electrons and lower the probability to recombine with holes. The depth of electron trap in anatase, rutile, and brookite are <0.1 eV, ∼0.9 eV, and ∼0.4 eV, respectively [163]. In comparison with brookite and rutile, anatase has the smallest depth of electron trap that indicates the presence of a larger number of free or shallowly trapped electrons. These electrons have higher reactivity than deeply trapped electrons. The lifetime of photogenerated electrons is prolonged. Consequently, anatase TiO_2_ has the highest photocatalytic reduction.

Furthermore, anatase TiO_2_ is known as an indirect band gap semiconductor. Looking into this, anatase has a longer lifetime of photogenerated electrons and holes as compared to brookite and rutile. Apart from this, the photogenerated electrons and holes in anatase can migrate easily from the innermost to the outermost surface of TiO_2_ due to lighter effective mass [164]. Additionally, the dominant (101) and (001) facets contribute to the high photocatalytic activity of anatase TiO_2_. The (001) facet contains rich under-bonded titanium atoms and a large Ti−O−Ti bond angle [155]. Anatase TiO_2_ usually exhibits higher photocatalytic activity compared to brookite and rutile TiO_2_ due to longer lifetime and lighter effective mass of photogenerated electrons and holes [164].

Rutile has octahedras sharing two edges to form a tetragonal crystalline structure which is identical to anatase [165]. In the rutile crystal structure, each octahedron is connected with eight similar octahedrons and this differentiates it from anatase where each octahedron is connected with ten similar octahedrons [155]. Rutile is thermodynamically stable in equivalent-sized TiO_2_ nanoparticles with sizes larger than 35 nm [161]. Rutile is stable at high temperature. This is proven where anatase and brookite transform into rutile at temperatures above 600–700 °C [166].

As explained earlier, rutile has the largest depth of electron trap, and this signifies that more electrons are deeply trapped at the defects of TiO_2_. The lifetime of holes is prolonged because the trapped electrons are unable to recombine with the holes [163]. Consequently, rutile TiO_2_ has the highest photocatalytic oxidation due to the higher availability of electron-scavenger. In addition, the smaller band gap energy of rutile TiO_2_ allows it to seize photons to produce electron and hole pairs that can be further used by anatase TiO_2_ [150]. It was reported that the heterojunctions of anatase/rutile TiO_2_ demonstrated higher photocatalytic activity than pure anatase or rutile TiO_2_ [167]. This was due to their matched band levels that restrain the recombination of photogenerated charge carriers.

In the brookite crystal structure, titanium atoms are located at the center, while oxygen atoms are located at each corner [168]. The TiO_6_ octahedra share three edges and form an orthorhombic crystalline structure [162]. Upon the precipitation in an acidic medium at low temperature, brookite is usually formed as the byproduct [152]. It is reported that brookite is thermodynamically stable in equivalent-sized TiO_2_ nanoparticles with sizes between 11 and 35 nm [161]. Similar to anatase, brookite will also transform to rutile upon calcination at high temperatures because it is metastable.

Brookite is a direct band gap semiconductor [164]. Based on the exposed surfaces of (201) and (210) facets, it can become oxidative and reductive surface facets, respectively. The moderate depth of the electron trap in brookite TiO_2_ resulted in both reactive electrons and holes. This explains the higher activity of brookite TiO_2_ in some photocatalytic reactions [163]. However, the detailed behavior of photogenerated electrons and holes in brookite TiO_2_ is not completely established yet [169]. Pure phase brookite is uncommon, difficult to prepare, and hence the discussion of brookite regarding its photocatalytic properties is limited.

### 4.2. Zinc Oxide (ZnO)

Besides of TiO_2_, ZnO is also a favorable semiconductor photocatalyst. The biocompatibility, excellent physicochemical stability, low production cost, great photocatalytic property, and high photosensitivity of ZnO promote its utilization in various energy conversion and photocatalytic activities [170,171]. In addition, ZnO displays strong luminescence in the green-white spectrum region which is appropriate for phosphor applications [172]. ZnO is an n-type semiconductor that has a direct and wide band gap in the UV region. The band gap energies of ZnO at low temperature and room temperature are 3.44 eV and 3.37 eV, respectively [172]. ZnO possesses a higher free-exciton binding energy (60 meV) than gallium nitride (25 meV). This shows that the excitonic emission in ZnO can occur at temperatures higher than room temperature. The photocatalytic degradation of ZnO is identical to TiO_2_ whereby the generated charge carriers produce free radicals to degrade organic pollutants under UV irradiation [173].

However, the application of pure ZnO is limited by some drawbacks. Firstly, ZnO requires expensive UV light for the wide band gap excitation. Secondly, the fast recombination of charge carriers in ZnO impedes the migration of charge carriers towards the outer surface of ZnO that leads to the retardation of degradation process [170]. Thirdly, the aggregation of ZnO particles reduces its dispersion and blocks the exposed active facets of ZnO [174]. Fourthly, ZnO faces photocorrosion under UV irradiation [175]. The ZnO powder dissolves in strong acidic and alkaline solutions, and eventually passivates to produce an inert outer layer of zinc hydroxide [174]. This determined that pure ZnO does not have sufficient activity under solar energy [170]. Lastly, the recovery of ZnO nanoparticles from post-treatment effluents is necessary owing to its toxicity. Upon respiratory and digestion uptake, the ZnO nanoparticles increase blood viscosity and heighten oxidative stress and cellular inflammatory response in the mammalian body [176].

ZnO is classified as an II-VI compound semiconductor whereby its ionicity remains at the cut-off point between covalent and ionic semiconductors [177]. Figure 6 shows that ZnO consists of three common crystal structures, which are rocksalt, zinc blende, and wurtzite. In these crystal structures, each anion is connected with four cations located at the edges of a tetrahedron, and vice versa [177]. The rocksalt structure of ZnO is considered rare and it can be obtained at high pressure [178]. Rocksalt has a cubic structure which is similar to zinc blende. The crystal structure of zinc blende is metastable. There are four atoms found in each unit cell of the zinc blende crystal structure. Each atom of group II is bonded with four atoms of group VI tetrahedrally and vice versa. It is noticeable that the tetrahedral coordination of zinc blende is identical to wurtzite.

Wurtzite is the most thermodynamically stable phase among the three crystal structures of ZnO. Wurtzite has a hexagonal structure at ambient conditions [178]. The stacking sequence of close-packed diatomic planes differentiates the crystal structure of wurtzite from zinc blende [177]. The crystal structure of zinc blende comprises triangularly arranged atoms in the close-packed (111) planes where the stacking order is in terms of “AaBbCcAaBbCc” along the (111) direction. On the other hand, the crystal structure of wurtzite comprised of triangularly arranged alternating biatomic close-packed (0001) planes where the stacking order is in terms of “AaBbAaBb” along the (0001) direction. The uppercase and lowercase letters indicate two distinct types of constituents. The low symmetry structure of wurtzite combines with a large electromechanical coupling to promote stronger piezoelectric and pyroelectric properties [172]. Thus, ZnO exhibits high piezoelectric constant, and it is commonly applied in sensors, transducers, and actuators.

### 4.3. Graphitic Carbon Nitride (g-C_3_N_4_)

In recent years, metal-free semiconductor photocatalysts received great attention due to their unique physicochemical properties. Carbon nitride semiconductors exist in different allotropes such as α-C_3_N_4_, β-C_3_N_4_, g-C_3_N_4_, and cubic-C_3_N_4_ [179]. Here, g-C_3_N_4_ is classified as the most stable allotrope under ambient conditions and this is proven with its popularity in photocatalytic applications [180]. This polymeric g-C_3_N_4_ has recently been reported as an easily available and simple photocatalyst for water splitting reactions without the presence of noble metals [181]. This outcome triggered the broad investigations of g-C_3_N_4_ in photocatalytic degradation of pollutants, carbon dioxide reduction, and hydrogen evolution [182]. The easy preparation, low cost, abundance in nature, and sustainable properties give rise to the widespread application of g-C_3_N_4_ [180].

Next, the tunable electronic band structure of g-C_3_N_4_ by nanomorphology or doping modifications improves its utilization in solar energy conversions (e.g., photoelectrochemical cells) [183]. During the photocatalytic process, the zero evolution of nitrogen gas (N_2_) determined the strong covalent bonding in pure g-C_3_N_4_ and exhibited high chemical stability of g-C_3_N_4_ [182]. In comparison with TiO_2_ and ZnO, g-C_3_N_4_ has a medium band gap of 2.7 eV, which enables it to absorb light at about 450–460 nm [184]. Under solar irradiation, the more negative conduction band potential (−1.3 V) and the more positive valence band potential (+1.4 V) of g-C_3_N_4_ versus NHE at pH 7 demonstrate its thermodynamic efficiency in participating redox reactions [155].

Although g-C_3_N_4_ contributes outstanding merits, it encounters some restrictions in photocatalytic activities. Firstly, the activity of g-C_3_N_4_ is often restricted by the fast recombination of charge carriers. Secondly, the small specific surface area (<10 m^2^/g) of g-C_3_N_4_ gives rise to its limited light-harvesting capability [185]. The reason is that a larger surface area indicates more reactive sites and higher light-harvesting capability. Thirdly, g-C_3_N_4_ is well known for its poor crystallinity and surface defects, which resulting in low conductivity in photocatalytic activities [182]. The crystallinity of g-C_3_N_4_ can be enhanced via the ionothermal approach [186]. Fourthly, the bulk g-C_3_N_4_ produced from conventional methods comprises multiple layers of two-dimensional (2D) counterparts [187]. The low specific surface area and irregular morphology of bulk g-C_3_N_4_ restrict its photocatalytic power [187]. Furthermore, pristine g-C_3_N_4_ also possesses high exciton binding energy as compared to inorganic photocatalysts [186]. Lastly, the organic semiconductor g-C_3_N_4_ encounters charge transport issues owing to the presence of grain boundaries [179].

Looking into the crystal structure of g-C_3_N_4_, it is an n-type and indirect semiconductor that exhibits 2D and 3D graphitic carbon [188]. The pyridinic and graphitic nitrogen in these structures are enriched with nitrogen content. g-C_3_N_4_ is identified as a conjugated polymeric arrangement and is composed of s-triazine or tri-s-triazine monomers interconnected via tertiary amines as shown in Figure 7 [179]. The combination of aromatic s-triazine rings and conjugated 2D polymer of s-triazine construct the π-conjugated planar layers which are similar to graphite [183]. Based on the density functional theory, it is determined that the carbon atoms are the ideal sites for the proton reduction to hydrogen, whereas the nitrogen atoms are the ideal sites for the water oxidation to oxygen [189]. The lone pair of nitrogen and the π bonding electronic states contribute to the stability of the lone pair state which results in the unique electronic structure of g-C_3_N_4_ [190].

Moreover, g-C_3_N_4_ can withstand thermal heat up to 600 °C in the air [183]. The high thermal and chemical stability of g-C_3_N_4_ is contributed by the high degree of condensation and the structure of tri-s-triazine ring. This polymeric g-C_3_N_4_ can be readily synthesized by thermal condensation with low-cost nitrogen-rich precursors (e.g., melamine, cyanamide, urea) [189]. The strong covalent bonds are responsible for the honeycomb arrangements of the atoms in the layers while the weak Van der Waals forces are responsible for stacking between the 2D sheets [179]. The Van der Waals forces also give rise to the good chemical stability of g-C_3_N_4_ in common solvents (e.g., water, alcohols, diethyl ether) [183].

### 4.4. Graphene

Geim and Novoselov were awarded the Nobel Prize for investigating the remarkable properties of graphene in 2010. The large theoretical specific surface area of graphene (2630 m^2^/g) gives rise to its strong adsorption capacity [191]. Graphene possesses good electron mobility that is 200 times higher than silica (1000 cm^2^/Vs) [192]. The fast-moving charge carriers do not scatter around under the presence of metal impurities while traveling through thousands of interatomic distances. This shows that graphene is suitable to be applied in high-speed operations due to power saving properties [192].

Furthermore, the cost of graphene production is low and it interacts strongly with transitional metals [193]. Graphene acts as a conductive support and an alternative electron sink whereby it accepts, stores, and shuttles photogenerated electrons [191]. This corresponds to the high conductivity and promising work function of graphene. Eventually, graphene exhibits high electrical conductivity (2000 S/m), high thermal conductivity (5000 W/mK), and good environmental compatibility [191]. The largest surface area, fastest electron mobility, highest conductivity, and outstanding electronic capability of graphene make it stand out from other materials like carbon nanotubes, graphite, and common metals.

As graphene is the mediator of electron transport, it prolongs the lifetime of photogenerated charge carriers and improves both the extraction and separation of charges [191]. This further elevates the photocatalytic activity of graphene-based composites. In addition, graphene serves as a building platform for the epitaxial growth of semiconductor nanostructures [171]. It also impedes the aggregation of these nanostructures which results in the improvements of exposed surface area and photocatalytic performance. Thus, graphene and its derivatives are widely applied in areas involving optical electronics, energy conversions, energy storages, photocatalysis, and photosensitizers [194].

Nevertheless, some limitations interrupt the application of graphene. Firstly, graphene faces the common problem of rapid recombination of electron–hole pairs. Secondly, graphene is a semi-metallic material and it has a zero band gap [195]. The zero band gap is formed from the contact between the π (bonding) and π* (antibonding) orbitals and the Brillouin zone corners. Alternatives like chemical doping could disrupt the lattice symmetry in graphene and eventually open a band gap with the help of foreign atoms. Thirdly, graphene-based powders tend to agglomerate easily and self-restack to form graphite [196]. The restacking phenomenon is irreversible and it is caused by the π–π stacking and Van der Waals forces located between the graphene sheets [197]. This causes the reduction of exposed surface area and ionic pathways [198]. Fourthly, pristine graphene is insoluble and it does not disperse uniformly in common solvents [194,199]. Lastly, graphene and its derivatives also encounter the familiar issues of recovery and separation after pollutant treatments.

Graphene consists of a monolayer of sp^2^-bonded carbon atoms [200]. It displays a hexagonal honeycomb lattice that is composed of two equivalent carbon sublattices [191]. Graphene is the basic structure of graphitic carbon allotropes as shown in Figure 8 [201]. The zero-dimensional fullerene is formed from wrapping a graphene sheet into a buckyball. The one-dimensional carbon nanotubes are formed from rolling graphene sheets into cylinder structures. The 3D graphite is formed from the stacking of graphene sheets as mentioned earlier. Monolayer and bi-layer graphene consists of one hole type and one electron type. The few layer graphene consists of three to nine graphene sheets whereas the multi-layer graphene is made up of ten or more graphene sheets [202]. The delocalized π bonds govern the graphene sheet-to-sheet interactions [203]. They allow the delocalization of π electrons along the basal plane.

Based on the structure of graphene, the π electrons found between the two adjacent carbons within its neighboring 2*p_z_* orbitals are related to the delocalized π and π* bands [191]. Here, the delocalized π band forms the highest occupied valence band while the π* band forms the lowest unoccupied conduction band with their respective π electrons. These two bands come into contact at the Dirac points, or, in other words, neutrality points. This indicated the semi-metallic properties and zero band gap of graphene. The π bonds are also responsible for the electrical conductivity of graphene [204]. The extended π–π conjugation contributes to the excellent strength of graphene. In terms of mechanical strength, graphene is reported with a Young’s modulus of 1 TPa [200]. Moreover, the surface of graphene is enriched with active functional groups like ketonic and quinonic [204]. This allows the functionalization of graphene and graphene-based materials because the functional groups can link easily with foreign molecules which results in the enhancement of photocatalytic performance.

## 5. Synthesizing Method of TiO_2_-Based Photocatalyst

Next, TiO_2_ can be prepared by various methods such as hydrothermal synthesis, sol-gel synthesis and chemical vapor deposition synthesis. These methods offer benefits like controlling the structural phases and stoichiometry, regulating the particle size, surface morphology, and homogeneity, promoting production of high purity nanomaterials, and are cost effective [205]. In the early stages of most synthesis, amorphous TiO_2_ with high surface area and porous structure is produced but these appealing properties tend to diminish after calcination at high temperatures where amorphous TiO_2_ is transformed into crystalline TiO_2_ [206]. Table 3 shows the photocatalytic degradation of pollutants using TiO_2_ catalysts prepared from hydrothermal method, sol-gel method, and chemical vapor deposition method.

### 5.1. Hydrothermal Synthesis

In general, hydrothermal synthesis is known as the crystallization above the room temperature and pressure through heterogeneous reactions in an aqueous/non-aqueous medium [223]. The hydrothermal synthesis is normally conducted within a sealed Teflon-lined stainless steel autoclave that can resist high temperature and pressure conditions. These conditions are applied to chemical substances that are hard to dissolve in solutions. It also can be assisted by ultrasonic irradiation, microwaves, and surface directing agents [224]. The crystallization in the hydrothermal synthesis involves two stages which are the nucleation and development of crystals [225].

In fact, hydrothermal synthesis produces a semiconductor, i.e., TiO_2_ that exhibits defect-free nano TiO_2_ crystals with large specific surface area, less particle agglomeration, narrow particle size distribution, anatase formation at temperature lower than 200 °C, and low energy consumption [206]. Experimental parameters such as hydrothermal temperature, time, pressure, solvent type, and titanium precursors can be controlled to modify the final product of TiO_2_ including crystallinity and porosity. However, the application of TiO_2_ to synthesize nanomaterials is limited, owing to the expansive sealed autoclave, inability to view the reaction in progress, and safety issues regarding the high temperature and pressure conditions [225].

In addition, the surface morphology of semiconductor photocatalysts could also affect their photocatalytic activities. For instance, Baral et al. [226] reported that the photocatalytic degradation efficiency of 20 mg/L dyes (Rhodamine B, Rhodamine—6G, Congo Red, Methyl Blue, and Methyl Orange) in the presence of α-manganese dioxide (MnO_2_) nanorods achieved 95–100% after 10 min under visible light irradiation (8 mW/cm^2^). The high photocatalytic activity of α-MnO_2_ nanorods was related to their one-dimensional morphology with a high aspect ratio and low photoluminescence intensity. The distinctive one-dimensional nanorod facilitated the separation of photogenerated charge carriers, as progressing charge separation along a single channel could lengthen the recombination time taken for electrons and holes.

Similar findings were reported on the performance of TiO_2_ photocatalysts with different morphologies fabricated via hydrothermal synthesis as shown in Table 3. TiO_2_ morphologies like nanoparticles, nanotubes, and nanosheets have been synthesized to test for environmental decontamination [223]. TiO_2_ nanoparticles with small primary particle size have large specific surface area and high pore volume that can improve the adsorption of contaminants, light-harvesting ability, and photocatalytic degradation. It is reported that the base solution employed in hydrothermal synthesis affects the morphology of one-dimensional TiO_2_ nanostructures. NaOH is commonly used as the solvent in hydrothermal synthesis to influence the crystalline phase and morphology of TiO_2_ nanostructures [224].

### 5.2. Sol-Gel Synthesis

Sol-gel synthesis is commonly used to prepare semiconductor metal oxide nanomaterials like TiO_2_ and ZnO. The sol-gel synthesis involves four processes which are hydrolysis, polycondensation, drying, and thermal decomposition [205]. In a typical sol-gel synthesis, the chemically active precursors are mixed uniformly in a liquid phase to undergo hydrolysis and condensation processes [227]. Here, water, alcohol, acid, and base can support the hydrolysis of the precursors [205]. A stable sol is formed after hydrolysis and condensation processes. The limiting factor of sol-gel synthesis is the necessary aging time for the catalyst. The range of aging time is 8–48 h [228]. Upon aging, a gel with 3D network structure is formed. After drying, thermal decomposition is carried out to remove the organic/inorganic precursors. In short, the sol-gel synthesis is focused on controllable hydrolysis, condensation of precursor type, and inorganic polymerization of the catalyst nanoparticles [205].

Next, the advantages of sol-gel synthesis are the formation of nanosized TiO_2_ with high purity at low temperature, possible stoichiometry-control process, good homogeneity, and fabrication of composite [229]. It is also simple, economical, and does not require any special and expensive instruments [227,230]. Despite that, the sol-gel synthesis may produce amorphous or weakly crystalline semiconductor metal oxide nanomaterials owing to the low fabrication temperature [205]. This points to the need for thermal decomposition such as annealing and calcination for further crystallization. Thermal annealing may induce hard aggregation and inter-particle sintering within the catalyst [231].

### 5.3. Chemical Vapor Deposition Synthesis

Chemical vapor deposition is commonly applied to produce semiconductors, i.e., TiO_2_ thin films. Generally, a thin and conformal solid film will be deposited on the surface of a substrate via chemical reactions of gaseous materials [232]. Here, single or multiple volatile precursors are introduced to the substrate at elevated temperature and pressure in an inert atmosphere [233]. The volatile precursors react or decompose onto the substrate to generate the ideal film thickness. The synthesis temperature of chemical vapor deposition method is usually applied at 200–1600 °C [232]. This method is still gaining momentum in the application of semiconductors, corrosion and wear-resistant coatings, monolithic components, etc. [234].

The chemical vapor deposition differs from physical vapor deposition where the films are formed under no chemical reactions in physical vapor deposition. It is reported that the durability, adhesion, and uniformity of film produced via chemical vapor deposition is better than physical vapor deposition [158]. Aging, drying, and reduction processes are also not required by applying chemical vapor deposition. Moreover, chemical vapor deposition gives advantages in terms of fabricating uniform and pure films, short fabrication time, allowing film formations on the inner pipe surfaces, and promoting good compatibility and adhesion [158,235]. Despite these advantages, drawbacks such as high deposition temperature, expensive vacuum systems, and safety issues with presence of corrosive gases hinder the application growth of chemical vapor deposition.

## 6. Fabrication and Performance of Cellulose Composite Catalysts

In general, the TiO_2_ photocatalyst encounters the main problem of recombination of the photogenerated electrons and holes. This causes a reduction in quantum yield and wastage of energy [147]. The modification of TiO_2_ photocatalyst with foreign atoms can alter the band gap and expand the adsorption range optically to improve the photocatalytic activity [236]. It is reported that the doping of TiO_2_ with metal and non-metal elements can increase charge separation to overcome the recombination problem mentioned earlier. The doping of TiO_2_ leads to a smaller photocatalyst loading required, lesser energy required, higher reusability of photocatalyst, shorter photocatalytic degradation time required, and higher photocatalytic activity [147].

The doping of TiO_2_ with metal elements includes noble metals, transition metals, and rare earth metals. Metal doping, or in other words, cation doping, mainly contributes towards the downward shift of the conduction band [237]. The conduction band of TiO_2_ contains the Ti 3d, 4s, and 4p orbitals whereby the Ti 3d orbitals govern the lower section of the conduction band. On the other hand, non-metal doping is an alternative approach to metal doping. Non-metal doping, or in other words, anion doping, mainly contributes towards the upward shift of the valence band. As a result, a new valence band is rebuilt to reduce the band gap. Other than doping, other modifications of photocatalyst involving the incorporation of cellulose and MOFs are further discussed later. Table 4 shows the photocatalytic degradation of dye by applying photocatalyst composites that incorporated cellulose, metal/non-metal doping, and MOFs.

Cellulose contain various functional groups like hydroxyl, carboxyl, and amino groups [256]. This also beneficial for the photocatalytic degradation of organic pollutants to less harmful substances. The cellulose-based adsorbents are also effective in the removal of pollutants including metal ions, dyes and pesticides as shown in Table 5. Besides, cellulose fibers are known for their large surface area, porous structure, low dielectric permittivity, biodegradability, and strong tensile strength [257]. Chen et al. [258] reported that the cellulose hydrogel promotes uniform dispersion of TiO_2_ nanoparticles and graphene oxide sheets while retaining the structure of the cellulose matrix. Cellulose could also act as a support to promote dye adsorption and delay the recombination of electron and holes that enhances the photocatalytic performance [259]. The significant findings on the reported cellulose-based catalyst composites were discussed as shown in Table 6.

### 6.1. Metal Doping

The purpose of metal doping is to minimize the band gap energy and to redshift the absorption from the UV region. The doping of metal ions induces an impurity level (intermediate energy level) that enhances the visible light absorption by taking the role of electron donor or acceptor [237]. Looking into this, metal doping introduces defects in the lattice structure of TiO_2_ or modifies the lattice degree and further prolongs the lifetime of TiO_2_ [281]. The metal ions can be easily doped into the lattice structure of TiO_2_ due to their close similarity in ionic radius [147]. According to previous research, the modification of photocatalysts (e.g., TiO_2_) with foreign atoms, i.e., metal or non-metal ion doping, can alter the band gap by expanding the light absorption range optically. The minimum energy required to excite an electron from the valence band to the conduction band is known as the band gap energy [282]. A lower band gap energy is advantageous for visible light absorption and photocatalytic activity [283]. When the band gap energy of photocatalysts is reduced, lesser energy from light irradiation is required to excite the electrons from the valence band to the conduction band of photocatalysts. Subsequently, this improves the redox reactions and enhances the photocatalytic activity of photocatalysts.

Despite these advantages, the required amount of metals must be minimized owing to their exceedingly high price [236]. Metal doping triggers localized d states in the band gap of TiO_2_ that becomes the recombination centers of electron–hole pairs [237]. When there are too many traps for charge carriers positioned on the surface of the catalyst bulk, the mobility of charge carriers will be lowered and they are more likely to recombine before reaching the surface [284]. This is often found in TiO_2_ photocatalysts loaded with metal content higher than 5 wt% [285]. Noble metals such as Ag and gold (Au) are toxic in nature which makes it necessary to use them in small amounts. Metal-doped TiO_2_ photocatalysts also encounter photocorrosion and thermal instability upon the photocatalytic degradation of organic pollutants [286].

Noble metals such as Ag, Au, platinum, and Pd, enhance the performance of TiO_2_ through charge transferring, electron trapping, and reduction of band gap energy [147]. Upon doping, it is preferable for the electrons to move from TiO_2_ to the noble metals and the movement tends to continue until the Fermi energy of both TiO_2_ and noble metals reaches an equilibrium state. A potential barrier is generated as a result of the distortion of band structure between TiO_2_ and the noble metals. This makes it difficult for electrons to move from TiO_2_ to the noble metal. As a consequence, there is an accumulation of negative electrons at the interface of TiO_2_. Looking into this, a positive charge layer is generated below the TiO_2_ surface to maintain electrical neutrality and this leads to the bending of a conduction band called Schottky barrier [284]. This barrier traps electrons effectively to prevent the electrons from flowing back to TiO_2_. The performance of noble-metal-doped TiO_2_ can be improved via surface plasmon resonance of noble metals, surface adsorption of TiO_2_ with organic compounds, and adjustment of Fermi level between TiO_2_ and noble metals to improve charge excitation and separation, and reduction of band gap energy [147].

Transition metals such as manganese, iron (Fe), and chromium, are also studied as metal dopants on TiO_2_. Transition metals that exhibit various oxidation states are capable of trapping electrons and impede their recombination [284]. For example, Fe has three ion forms which are Fe^2+^, Fe^3+^, and Fe^4+^. Nevertheless, it is reported that the transition metal is likely to cause thermal instability in anatase TiO_2_. Rare earth metals including yttrium and scandium, are investigated as metal dopants on TiO_2_ photocatalyst. They consist of incomplete 4f and unoccupied 5d orbitals that can be incorporated into TiO_2_ [147]. The presence of rare earth metal−O−Ti bonds on the surface of TiO_2_ photocatalyst impedes the growth of nanocrystalline TiO_2_.

### 6.2. Non-Metal Doping

As stated earlier, the non-metal doping of TiO_2_ shifts the valence band upwards. Unlike metal doping, non-metal doping does not affect the shifting of conduction band [237]. An intermediate energy level is formed between the conduction bands of TiO_2_ and non-metal dopants. Non-metal doping inhibits the recombination of electron–hole pairs, improves the redox potential of OH^•^, and finally enhances the quantum efficiency of TiO_2_ [147]. N, fluorine, S, carbon (C), boron, and phosphorus are the common non-metal dopants on the TiO_2_ photocatalyst.

Currently, N is the most studied dopant for non-metal doping of TiO_2_ due to good stability, low ionization energy, and comparable atomic size [281,286]. Typically, N can be easily incorporated into TiO_2_ in terms of substitutional and interstitial. The oxygen vacancies are the defects that can trap electrons at the defect sites [284]. Similar to metal doping, N doping also acts as an electron trapping center to decrease the recombination rate of electron–hole pairs. However, the main drawback of non-metal doping is the reduction of non-metal dopants during the annealing process [287]. The reduction of non-metal dopant in the resultant doped TiO_2_ photocatalyst lowers its performance. Co-doping of TiO_2_ is investigated to overcome the single non-metal doped TiO_2_ photocatalysts.

### 6.3. Metal–Organic Frameworks (MOFs)

MOFs are known as new porous coordination polymers [288]. They consist of metal ions/clusters linked by multi-dentate organic linkers [289]. Their advantages, including large surface area, tunable pore size and structures, and being rich with active sites have been applied in catalysis and wastewater treatments. MOF-199, MIL-100, MIL-125, and ZIF-8 are some of the popular MOFs. Due to the large surface area and diverse functionality, MOFs act as appealing adsorbents and can be modified to obtain higher selective binding affinities for the targeted adsorbates [290]. Table 5 shows the removal of pollutants by cellulose/MOF composites via adsorption process.

Moreover, the performance of MOFs photocatalysts are similar to common semiconductors (TiO_2_ and ZnO) and they are widely used in organic pollutant degradation and Cr(VI) reduction [289]. The tunable metal ions/clusters and organic linkers act as antennas to harvest light and generate electron–hole pairs for photocatalysis. Nevertheless, the excited electrons from highest occupied molecular orbital will travel to lowest unoccupied molecular orbital during the photocatalytic process in the presence of MOFs [289]. This terminology is different from the common semiconductors whereby the excited electrons will travel from the valence band to the conduction band.

## 7. Challenges and Future Perspective

Agricultural wastes have some agronomic benefits over wood as a biomass resource including shorter growth cycles, high biomass yield, and high carbohydrate content [291]. Nevertheless, the average increase rate of agricultural waste generated is 5–10% annually [292]. This rapid rate contributes to negative environmental impacts. The biomass wastes are mostly left in fields for natural decomposition, discarded at landfills, or incinerated for cooking process, drying process, and charcoal production [293]. This could give rise to GHG emissions and air quality deterioration. The conversion of biomass wastes into value-added materials supports the purpose of a circular economy and mitigates the biomass disposal problem.

Next, agricultural biomass contains high cellulose content. This potential cellulose source has drawn the attention of researchers to fabricate and develop biomass-derived cellulose and its derivatives over the years. Cellulose and its derivatives are commonly applied in industries such as paper, textiles, pharmaceuticals, and water treatment [294]. The common challenge faced during cellulose extraction from biomass is biomass recalcitrance which involves the resistance of the lignocellulosic biomass structure towards microbial or enzymatic degradation [295]. The removal of lignin and hemicellulose is necessary to obtain cellulosic materials with high purity. However, the harsh chemical treatments on biomass could lead to the formation of partially degraded cellulose and the generation of toxic effluents [294]. A greener approach should be taken into consideration for the fabrication of biomass-derived cellulosic products. Chlorine-based compounds are widely used in the bleaching and synthesis stages, and yet the effluents containing chlorine give out harmful impacts to the environment [296]. Chlorine-free preparation routes for biomass-derived cellulose products should be emphasized for future commercial-scale applications.

Semiconductor photocatalysts such as TiO_2_, ZnO, g-C_3_N_4_, and graphene are widely used in the degradation of organic pollutants due to their high chemical stability and high thermal stability. The bulk forms of metal oxides (e.g., TiO_2_, ZnO) are safer as compared to their nanoparticulate forms because the dispersion of nanoparticles with high oxidation potential in aqueous medium could attribute to higher toxicity [297]. The interesting features of cellulose make it a suitable candidate to fabricate cellulose-semiconductor photocatalyst composites. Besides modifications involving metal/non-metal doping, the incorporation of cellulose could also address the drawbacks of semiconductor photocatalysts alone. Cellulose could promote the uniform dispersion of nanoparticles, increase specific surface area, improve adsorption ability, inhibit recombination of charge carriers, and reduce band gap energy. Biomass-derived cellulose are usually considered to be good candidates as host materials of nanomaterials because they can improve the stability, retain the special morphology, and control the growth of nanoparticles. The excellent properties in terms of electrical, magnetic, and optical in inorganic nanoparticles, can be preserved in the polymer matrix. These potential characteristics are beneficial for the photocatalytic degradation of organic pollutants.

In recent years, numerous novel cellulose-semiconductor photocatalyst composites are being reported with high catalytic performances in water treatment applications. This has proven the importance and practical feasibility of cellulose in the development of photocatalyst nanomaterials. Other than the reported degradation mechanism of cellulose-based photocatalysts, a thorough investigation of the photostability and long usage period of cellulose should be taken into account. In the case of photocatalytic degradation, the photocatalysts are exposed to light irradiation for a long time period. Post-treatment characterization studies on the cellulose-based photocatalysts are encouraged to observe and determine any signs of photodegradation or altered structure of cellulose within the composite. A better understanding of this crucial feature could help develop sustainable and long-lasting cellulose-semiconductor photocatalysts in future research works.

## 8. Conclusions

Biopolymers, such as cellulose and chitosan, have gained tremendous attraction in environmental and biological applications due to their abundance and multi-functionality. The appealing characteristics of biopolymers including large surface area, non-toxic, anti-microbial activity, and biocompatibility have led to the applications in food packaging, wastewater treatment, and biofuel. Since cellulose is the main component of biomass, the advancements in cellulose isolation methods have enabled the utilization of biomass-derived cellulose in environmental applications. The hydroxyl groups found in the chemical structure of cellulose are responsible for controlling the crystalline packing, physical properties, and chemical reactivity of cellulose. This is beneficial for the functionalization of cellulose materials with foreign substances.

The promising aspects of cellulose, including that they are low cost, abundant, and environmentally friendly, are suitable for the fabrication of cellulose hybrid composite photocatalysts. The incorporation of cellulose could overcome the limitations of semiconductor photocatalysts. The large specific surface area and porous structure of cellulose materials could promote dye adsorption on the surface of photocatalysts. The introduction of cellulose minimizes the aggregation of semiconductor particles, impedes the recombination of charge carriers, and reduces the band gap energy of semiconductor photocatalysts. The generation of large amounts of reactive radicals enhances the photocatalytic performance. These notable characteristics are favorable for the photocatalytic degradation of organic dye. In addition, the modifications such as doping, and MOFs also enhanced the catalytic performance of cellulose composite photocatalysts. The high performance and sustainability of cellulose hybrid composite photocatalysts have proven the advantage of cellulose and showed a more promising advancement in the development of photocatalysts.

Agricultural biomass has a high cellulose content, making it a potential cellulose source. However, biomass recalcitrance limits and hinders the production of biomass-derived cellulose. High purity cellulose can be extracted from biomass by highly toxic chemical processes but doing so would also partially degrade the cellulose and produce harmful effluent or any byproducts. Therefore, a more environmentally friendly strategy, such as preparation routes for cellulose obtained from biomass without the use of chlorine, could ensure the discharge of less hazardous effluents into the environment. On the other hand, the high feasibility and significance of cellulose in the development of various photocatalysts has been reported over the years. Nevertheless, research on the photostability and long usage period of cellulose is limited. These critical aspects can be investigated through post-treatment characterization studies on the cellulose-based photocatalysts. It also serves as supporting information for the future development of robust cellulose-semiconductor photocatalysts.

## Figures and Tables

**Figure 1 polymers-14-05244-f001:**
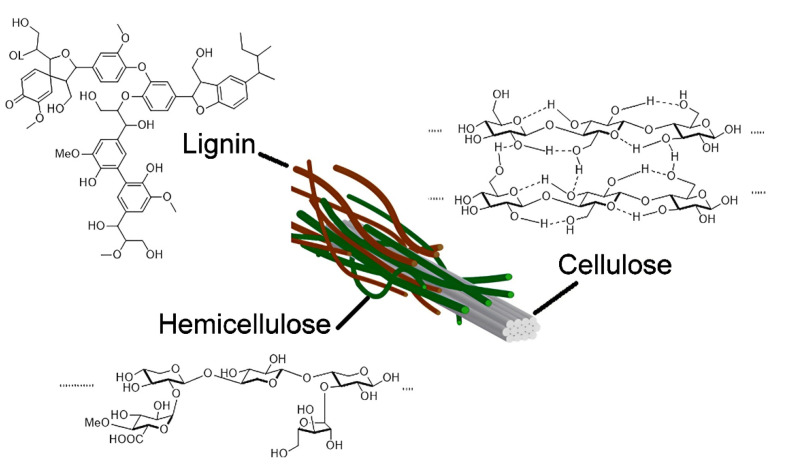
Cellulose, hemicellulose, and lignin contained inside biomass. Reproduced with permission from [14].

**Figure 2 polymers-14-05244-f002:**
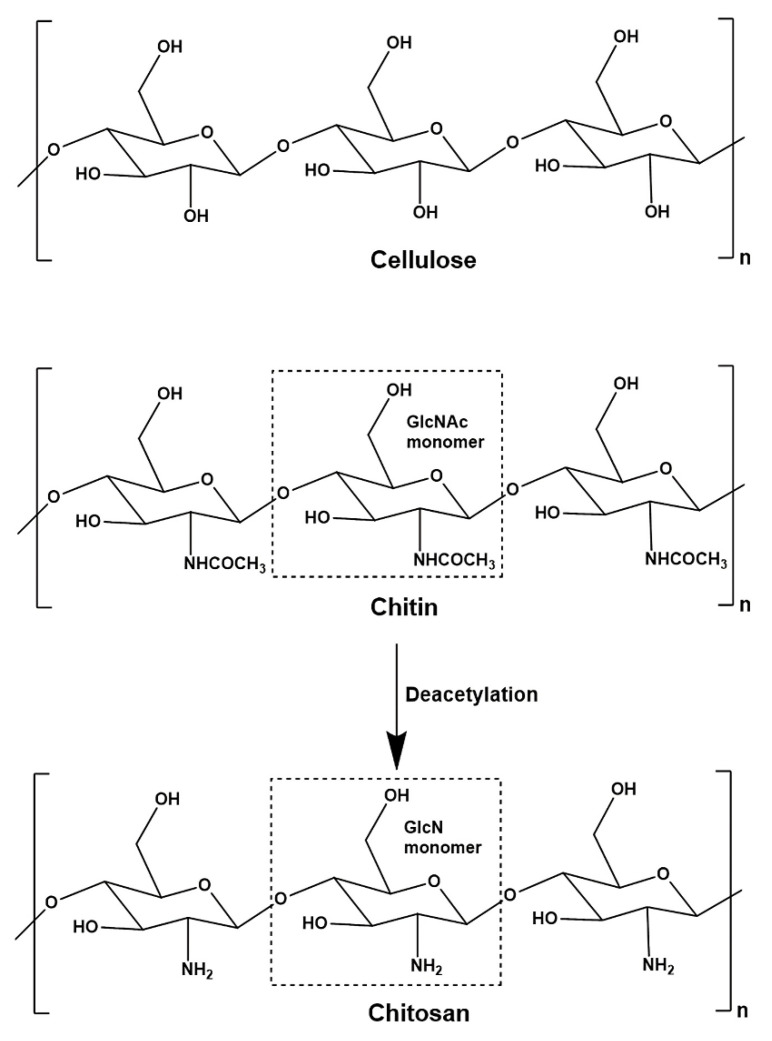
Structure comparison between cellulose, chitin, and chitosan. No special permission is required to reuse all or part of an article published by MDPI [64].

**Figure 3 polymers-14-05244-f003:**
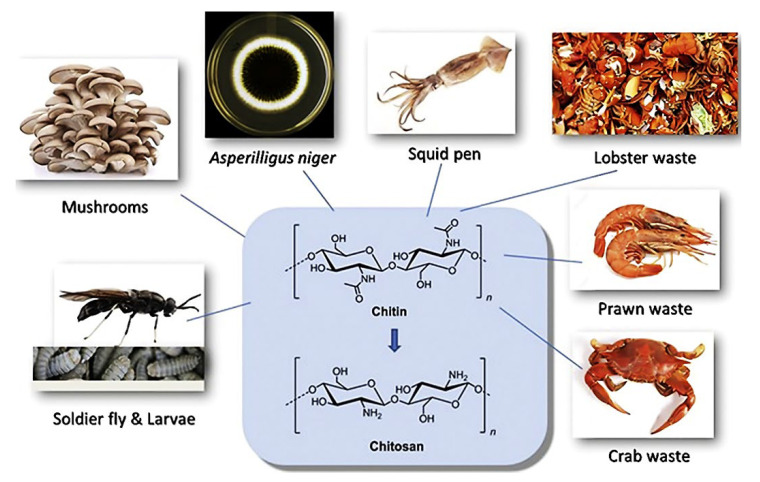
The chitin sources. Reproduced with permission from [66].

**Figure 4 polymers-14-05244-f004:**
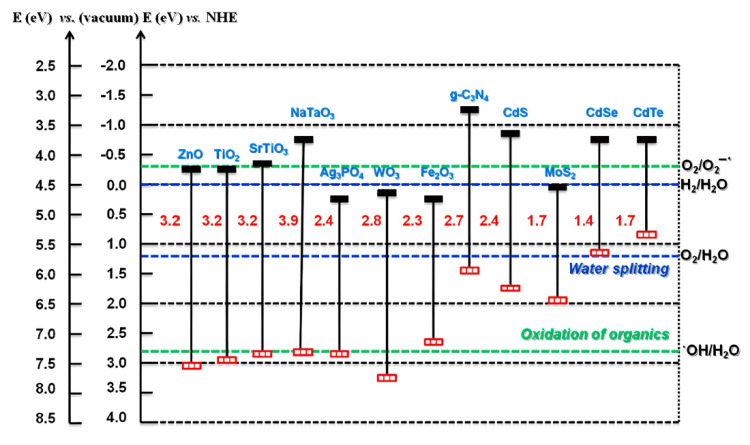
Band gap energy of various semiconductor photocatalysts. No special permission is required to reuse all or part of article published by MDPI [159].

**Figure 5 polymers-14-05244-f005:**
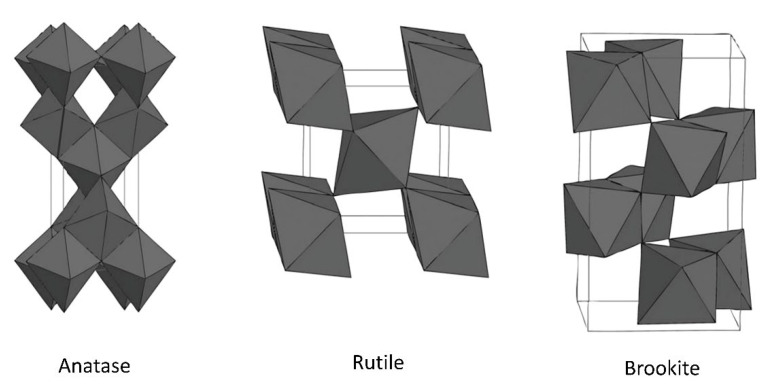
Crystal structures of TiO_2_. Reproduced with permission from [152].

**Figure 6 polymers-14-05244-f006:**
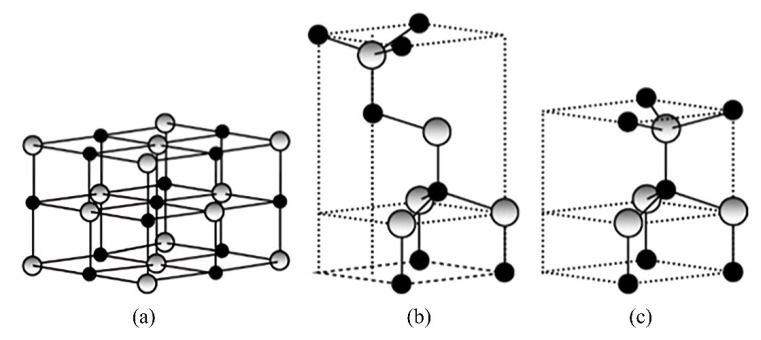
Crystal structures of ZnO: (**a**) rocksalt, (**b**) zinc blende, and (**c**) wurtzite. Reproduced with permission from [178].

**Figure 7 polymers-14-05244-f007:**
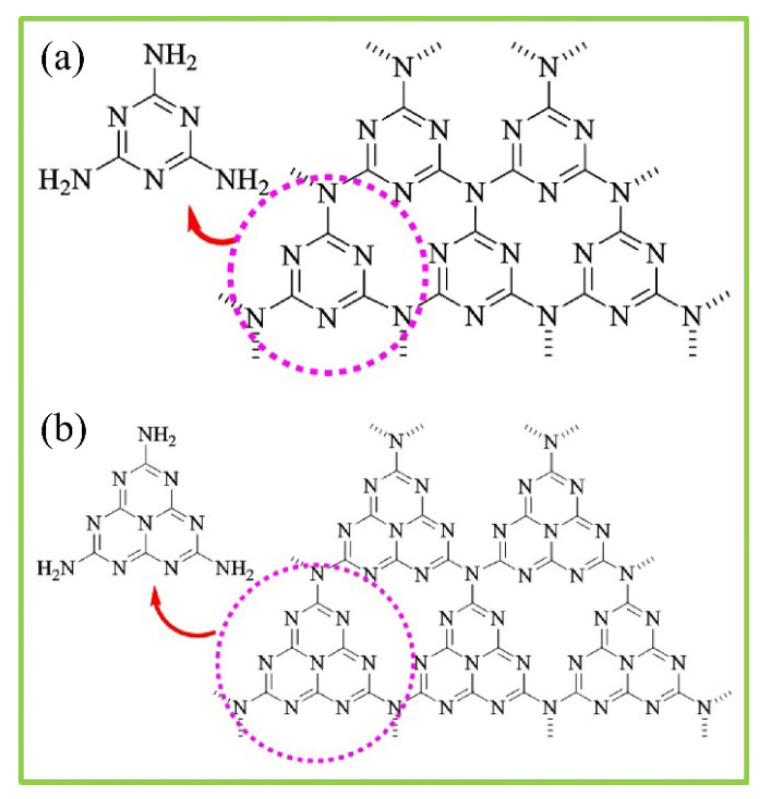
Structures of g-C_3_N_4_: (**a**) s-triazine based, and (**b**) tri-s-triazine based. Reproduced with permission from [184].

**Figure 8 polymers-14-05244-f008:**
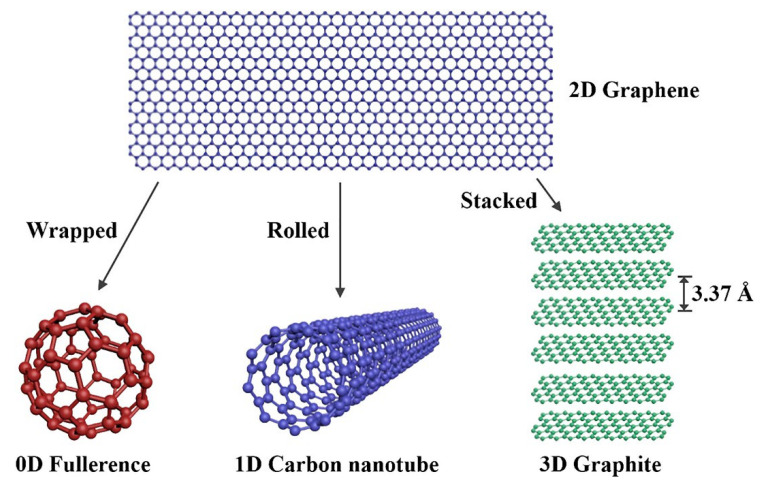
Structures of graphene and its graphitic carbon allotropes. Reproduced with permission from [201].

**Table 1 polymers-14-05244-t001:** Preparation of MCC and NCC from various sources using different acid hydrolysis methods.

Cellulose Form	Cellulose Source	Acid Hydrolysis Method	MCC/NCC Yield (%)	α-Cellulose Content (%)	Crystallinity (%)	Thermal Stability (°C)	Ref
MCC	*Ensete glaucum* (Roxb.) Cheesman	2.5 M HCl at 105 °C for 15 min	33	99	53.41	-	[85]
	Sweet sorghum	7 wt% HCl at 40 °C for 90 min	81.8	93.2	75.19	-	[86]
	Kans grass	5% (w/w) H_2_SO_4_ at 50 °C for 120 min	83	83.33	74.06	338	[87]
	Date seeds	2.5 M HCl at 105 °C for 45 min	12.51	-	70	352.52	[88]
	Conocarpus fiber	2.5 M HCl at 80 °C for 30 min	27	-	75.7	408.5	[89]
NCC	Rice husk	4 M H_2_SO_4_ at 60 °C for 60 min	95	95	65	-	[90]
	OPEFB	3 M HCl at 80 °C 120 min	21	94.6	65	358.5	[91]
	Jackfruit peel	65% (w/w) H_2_SO_4_ at 37 °C for 60 min	7	20.08	83.42	-	[92]
	Olive fiber	35 wt% H_2_SO_4_ at 40−50 °C for 60 min	16.4	86.2	83.1	363.8	[93]
	Rice husk	64 wt% H_2_SO_4_ at 45 °C for 30 min	35–37	-	82.8	286	[94]

**Table 2 polymers-14-05244-t002:** Extraction of cellulose from lignocellulosic biomass using different combined treatment methods.

Cellulose Source	Cellulose Extraction Method	Cellulose Content (%)	Crystallinity (%)	Thermal Stability (°C)	Ref
Rice straw	1.25% acidified NaClO_2_ at 75 °C for 1 h 5 wt% KOH at room temperature for 16 h followed by 90 °C for 2 h	88.5	58.12	358	[98]
Oil palm frond	15 wt% NaOH at 150 °C and 7 bar for 1 h 10% H_2_O_2_ at 90–100 °C for 1 h	91.33	77.78	366.8	[99]
Sugarcane bagasse	10% (*v*/*v*) H_2_SO_4_ at 100 °C for 1 h 5% (*m*/*v*) NaOH at 100 °C for 1 h 5% (*v*/*v*) H_2_O_2_ and 0.1% MgSO_4_ (in polypropylene bags) at 70 °C for 1 h	89.12	56.19	360	[100]
Wheat straw	0.5 mL CH_3_COOH and 1 g NaClO_2_ in 80 mL water at reflux (oil bath at 80 °C) for 4 h 17.5% (*w*/*v*) NaOH at room temperature for 30 min	81.4	66.6	385	[101]
	1% (*w*/*v*) NaOH and 20% (*w*/*v*) H_2_O_2_ at 121 °C for 35 min	79	66.87	360	
*Agave gigantea*	5% (*w*/*v*) NaOH at 80 °C for 2 h NaOH/CH_3_COOH/water (27 g/75 mL/1 L) and 1.7 wt% NaClO_2_ at 80 °C for 1 h	89.39	70.94	362.59	[102]

**Table 3 polymers-14-05244-t003:** Performance of TiO_2_ prepared from different synthesis methods in pollutant treatments.

Synthesis Method	Sample	Conditions	Pollutant Type	Treatment Time (min)	Degradation (%)	Ref
Hydrothermal	TiO_2_ nanowires	(1) Titanium (IV) butoxide (TBT), ethanol, and 10 M NaOH within autoclave at 180 °C for 24 h (2) Calcined at 650 °C for 2 h	10 mg/L Rhodamine B	60	100	[207]
	TiO_2_ nanorods	(1) Titanium (IV) isopropoxide (TTIP) and NaOH within autoclave at 180 °C for 24 h (2) Annealed at 400 °C for 5 h	10 mg/L Methyl Orange	150	51	[208]
	TiO_2_ nanorod arrays	(1) TBT, HCl, and fluorine-doped tin oxide substrate within autoclave at 170 °C for 5 h (2) Calcined at 450 °C for 2 h	5 mg/L Bisphenol A	180	49	[209]
	TiO_2_ nanotubes	(1) Titanium sulfate and urea within reaction vessel at 220 °C for 12 h (2) Resultant TiO_2_ powder and 10 M NaOH within Teflon at 150 °C for 12 h	20 mg/L Tetracycline Hydro-chloride	60	23	[210]
	TiO_2_ nanosheets	(1) TBT and hydrofluoric acid within autoclave at 180 °C for 24 h	10 mg/L Rhodamine B	80	36.5	[211]
Sol-gel	TiO_2_	(1) Deionized water, ethanol, CH_3_COOH, and sodium dodecyl sulfate stirred for 30 min (2) Resultant mixture and TTIP stirred at 25 °C for 24 h followed by at 60 °C for 24 h (3) Calcined at 450 °C for 4 h	20 mg/L Methylene Blue	30	99	[212]
	TiO_2_	(1) Titanium (IV) chloride (TiCl_4_) and H_2_SO_4_, followed by ammonia to achieve pH 7–8 (2) Annealed at 500 °C	10 mg/L Thymol	120	9.65	[213]
	TiO_2_	(1) TBT, CH_3_COOH, and water stirred at 80 °C (2) Annealed at 500 °C for 5 h	0.03 mg/L Methylene Blue	120	96	[214]
	TiO_2_	(1) TiCl_4_, deionized double distilled water, and 30% ammonium hydroxide stand for 1 h	20 mg/L Alizarin	60	71	[215]
	TiO_2_	(1) TTIP, isopropanol, acetylacetone, and water followed by HCl to achieve pH 3, the resultant mixture A stirred for 4 h and aged for 24 h (2) Isopropanol and water followed by HCl to achieve pH 3, the resultant mixture B refluxed at 60 °C for 24 h and aged for 24 h (3) Resultant mixture A and B stirred for 1 h, heated at 60 °C for 2 h, and followed by thermal treatment at 500 °C for 1 h	50 mg/L Quinoline	180	51	[216]
Chemical vapor deposition	TiO_2_	(1) Reactive sputtering using a titanium (Ti) target with gaseous mixture of argon (Ar) and O_2_ (2) Resultant TiO_2_ deposited on polyvinylidene difluoride membrane for 1 h	4.1587 mg/L Methylene Blue	240	92	[217]
	TiO_2_	(1) TTIP as precursor and Ar as carrier gas (2) Resultant TiO_2_ deposited on alumina balls at 500 °C for 1 h	246.22 mg/L Nitro-benzene	100	99	[218]
			20 mg/L Cimetidine	180	98.2	[219]
	TiO_2_/ clay	(1) TTIP as precursor and N_2_ as carrier gas (2) Resultant TiO_2_ deposited on ion-exchanged Na^+^–clay at 600 °C for 5 h	75 mg/L Methyl Green	60	87.2	[220]
	Zinc ferrite@ TiO_2_	(1) TTIP as precursor, Ar as carrier gas, and O_2_ as oxidant gas (2) Resultant TiO_2_ deposited on zinc ferrite nanofibers at 150 °C for 1 h (3) Calcined at 550 °C for 3 h	20 mg/L Methylene Blue	180	98	[221]
	Sulfur (S)-doped TiO_2_	(1) TTIP as precursor and N_2_ as carrier and purge gas (2) Resultant TiO_2_ deposited on borosilicate glass substrate at 400 °C and 50 mbar (3) Resultant TiO_2_ film and hydrogen-2v.% hydrogen sulfide at 50 °C for 60 min	5 mg/L Methyl Orange	300	72.1	[222]

**Table 4 polymers-14-05244-t004:** Performance of cellulose composites, cellulose/metal doped composites, cellulose/non-metal doped composites, and cellulose/MOF composites as photocatalysts in the degradation of organic dyes.

Composite Type	Sample	Dye Type	Catalyst Loading (g/L)	Power (W)	Treatment Time (min)	Degradation (%)	Ref
Cellulose	ZnO/CNF	5 mg/L Methylene Blue	2	9	30	96	[238]
	Chromium oxide/cellulose	10 mg/L Crystal Violet	0.1	25	40	99.65	[239]
	Bismuth oxybromide/cellulose-derived carbon nanofibers	10 mg/L Rhodamine B	0.5	200	60	100	[240]
	Beta-iron oxyhydroxide (β-FeOOH)/cellulose	10 mg/L Methylene Blue	1.3	300	30	99.89	[241]
	Bacterial cellulose (BC)/polydopamine/TiO_2_	20 mg/L Methylene Blue	0.6	500	20	99.5	[242]
		20 mg/L Methyl Orange			30	95.1	
		20 mg/L Rhodamine B			60	100	
Cellulose/ metal doped	Copper (Cu)@cuprous oxide/reduced graphene oxide/cellulose	10 mg/L Methyl Orange	3	350	120	92.8	[243]
	Aluminum-doped ZnO/cellulose	10 mg/L Methyl Orange	3	500	360	89.9	[244]
	Cu-CNF/TiO_2_	50 mg/L Reactive Brilliant Red K-2BP	0.6	300	120	96.57	[245]
		50 mg/L Cationic Red X-GRL				99.73	
	Ag-cadmium selenide (CdSe)/graphene oxide@cellulose acetate	5 mg/L Malachite Green	4	300	25	97	[246]
	CNF-Indium-doped Mo(O,S)_2_	10 mg/L Methylene Blue	1	150	30	100	[247]
		10 mg/L Methyl Orange			240	90	
		10 mg/L Rhodamine B			240	100	
Cellulose/ non-metal doped	Regenerated cellulose membrane-templated C-doped/core shell TiO_2_	10 mg/L Methylene Blue	0.05	300	120	90.1	[248]
	Nitrogen (N)-doped BC/TiO_2_	10 mg/L Methyl Blue	0.5	300	30	100	[249]
		10 mg/L Rhodamine B	0.5		35		
		20 mg/L Methyl Orange	1		15		
	N and S doped carbon dot CNF	5 mg/L Methylene Blue	-	1000	25	98	[250]
	C–TiO_2_/cellulose acetate	20 mg/L Reactive Red-195	5	125	60	99.15	[251]
Cellulose/ MOF	Europium-MOF@viscose fabric	20 mg/L Rhodamine B	-	500	120	97	[252]
	Phosphotungstic acid/zeolitic imidazolate framework(ZIF)-8@cellulose	10 mg/L Methylene Blue	0.6	-	30	99.8	[253]
	Ag@silver chloride@Material Institute Lavoisier(MIL)-100(Fe)/cotton fabric	20 mg/L Methylene Blue	0.125–0.15	500	40	100	[254]
		20 mg/L Rhodamine B					
	β-FeOOH@MIL-100(Fe)/cellulose/polyvinyl pyrrolidone	20 mg/L Methylene Blue	0.125	500	20	99.4	[255]

**Table 5 polymers-14-05244-t005:** Performance of cellulose and cellulose/MOF adsorbents on the removal of pollutants.

Composite Type	Sample	Adsorbent Dosage (g/L)	Pollutant Type	pH	Contact Time (min)	Adsorption Capacity (mg/g)	Ref
Cellulose	Cellulose-*g*-hydroxyethyl methacrylate-*co*-glycidyl methacrylate	1	20 mg/L Malachite Green	7	360	24.88	[260]
			20 mg/L Crystal Violet			19.51	
	Cellulose-*g*-2-acrylamido-2-methylpropane sulfonic acid-*co*-glycidyl methacrylate	1	100 mg/L Cu^2+^	6	120	78.247	[261]
			100 mg/L Ni^2+^	5		69.061	
	Magnetite-functionalized NCCs/starch-g-(2-acrylamido-2methyl propane sulfonate-co-acrylic acid)	1	1000 mg/L Crystal Violet	9	120	2500	[262]
			1000 mg/L Methylene Blue			1428.6	
	Cadmium sulfide@silanized CNF	1	9.597 mg/L Methylene Blue	11	360	26.66	[263]
			10.5255 mg/L Safranin-O	11		17.857	
			28.0472 mg/L Chlorpyrifos	3		86.9565	
	Dual cross-linked—alginate/treated biomass bead	0.4	210 mg/L Pb^2+^	5	120	206.75	[264]
Cellulose/ MOF	Cellulose acetate/MOF-derived porous carbon	0.1	50 mg/L Methylene Blue	11	360	41.36	[265]
	MOF-199/cellulose/chitosan	0.3	50 mg/L Methylene Blue	7.5	1440	161.7	[266]
	BC@ZIF-8	0.28	180 mg/L UO_2_^2+^	3	120	387.13	[267]
	MOF-199@cellulose acetate	0.8	20 mg/L Dimethoate	7	360	321.9	[268]
	Waste paper@polystyrene sulfonate@Cu-MOF	1	150 mg/L Li^+^	9	360	9.69	[269]

**Table 6 polymers-14-05244-t006:** Significant findings on the performance of cellulose-based photocatalysts in the degradation of pollutants.

Composite Type	Sample	Significant Findings	Ref
Cellulose	ZnO/NCC	The incorporation of NCCs increased the specific surface area of bare ZnO from 13.014 m^2^/g to 32.202 m^2^/g. The photocatalytic degradation of Methylene Blue (10 mg/L) in the presence of ZnO and ZnO/NCC reached 65.87% and 88.62%, respectively, after 120 min under solar light irradiation.	[270]
	g-C_3_N_4_−CNF	The band gap energy of g-C_3_N_4_−CNF nanocomposite foam (2.7 eV) was lower than bare g-C_3_N_4_ (2.9 eV). The photocatalytic degradation of Rhodamine B (5 mg/L) in the presence of g-C_3_N_4_ and g-C_3_N_4_−CNF achieved 16% and 47%, respectively, after 6 h of visible light irradiation in the absence of agitation.	[271]
	Cellulose/bismuth vanadate	The addition of cellulose to bismuth vanadate reduced its crystallite size and band gap energy from 22.03 nm to 6.55 nm and from 2.93 eV to 2.44 eV, respectively. The photocatalytic degradation of Methyl Orange (3.2734 mg/L) in the presence of cellulose/bismuth vanadate reached 80% after 60 min under visible light irradiation (200 W).	[259]
Cellulose/ metal doped	Molybdenum-doped TiO_2_ films templated by NCCs	The specific surface area of NCCs-templated TiO_2_ films doped with molybdenum (149 m^2^/g) was larger than NCCs-templated TiO_2_ films (103 m^2^/g). The photocatalytic degradation of trichloroethylene (200 ppmv) in the presence of NCCs-templated TiO_2_ film doped with 5 at% molybdenum achieved 100% after 30 min under UV irradiation (15 W).	[272]
	Iron (II, III) oxide(Fe_3_O_4_)/praseodymium-bismuth oxychloride (BiOCl)/cellulose	The specific surface areas of Fe_3_O_4_/BiOCl, Fe_3_O_4_/praseodymium-BiOCl, and Fe_3_O_4_/praseodymium-BiOCl/cellulose were 3.5315 m^2^/g, 4.2115 m^2^/g, and 17.6303 m^2^/g, respectively. The photocatalytic degradation of Rhodamine B (10 mg/L) in the presence of Fe_3_O_4_/praseodymium-BiOCl/cellulose accomplished 99.8% after 120 min under visible light irradiation (300 W).	[273]
	Ag/TiO_2_@ cellulose-derived carbon beads	The band gap energy of Ag/TiO_2_@cellulose-derived carbon beads (2.21 eV) was lower than pure TiO_2_ (3.08 eV). The photocatalytic degradation of ceftriaxone sodium (25 mg/L) in the presence of Ag/TiO_2_@cellulose-derived carbon beads reached 91.92% after 270 min under visible light irradiation (500 W).	[274]
Cellulose/ non-metal doped	Lanthanum–N–TiO_2_–cellulose/silicon dioxide (SiO_2_)	The band gap energy of TiO_2_ was reduced from 3.59 eV to 2.81 eV. The photocatalytic degradation of crude oil solution (20 mg/L) in the presence of lanthanum–N–TiO_2_–cellulose/SiO_2_ achieved 92% after 180 min under sunlight.	[275]
	N-TiO_2_/C	The photocatalytic degradation of 4-nitrophenol (10 mg/L) in the presence of N-TiO_2_/C reached 80% after 420 min under visible light irradiation (24 W). The effect of N doping improved the photocatalytic activity of TiO_2_/C by 26%.	[276]
	TEMPO-oxidized cellulose-ZnO	The specific surface area and pore volume of TEMPO-oxidized cellulose-ZnO (20.1 m^2^/g and 0.09 cm^3^/g) were higher than bare ZnO (9.1 m^2^/g and 0.05 cm^3^/g). The band gap energy of ZnO was reduced from 3.15 eV to 3.11 eV. The photocatalytic degradation of Methyl Orange (5 mg/L) in the presence of TEMPO-oxidized cellulose-ZnO obtained 95.4% after 120 min under UV irradiation (6 W). There was an improvement of 29.5% in the photocatalytic activity of TEMPO-oxidized cellulose-ZnO as compared to bare ZnO.	[277]
Cellulose/ MOF	TiO_2_/ magnetic-MIL-101(chromium)	The band gap energy of TiO_2_/magnetic-MIL-101(chromium) (1.61 eV) was lower than pure TiO_2_ (3.1 eV). The photocatalytic degradation of Acid Red 1 (20 mg/L) in the presence of TiO_2_/magnetic-MIL-101(chromium) achieved 90% after 60 min under visible light irradiation (500 W).	[278]
	(Ag & palladium (Pd))@MIL-125-NH_2_@cellulose acetate	The band gap energy of MIL-125-NH_2_@cellulose acetate, Ag@MIL-125-NH_2_@cellulose acetate, and Pd@MIL-125-NH_2_@cellulose acetate films were 2.53, 2.38, and 1.99 eV, respectively. The photocatalytic reduction of 2-nitrophenol (25 mg/L) in the presence of Ag-doped and Pd-doped MIL-125-NH_2_@cellulose acetate films reached 87.4% and 95.1%, respectively, after 120 min under visible light irradiation (12 W). The metal-doped MIL-125-NH_2_@cellulose acetate films doubly increased the photocatalytic reduction of 2-nitrophenol in 120 min as compared to MIL-125-NH_2_@cellulose acetate alone.	[279]
	Cellulose acetate@Ti-MIL-NH_2_	The insertion of Ti-MIL-NH_2_ within the cellulose acetate film increased the surface area from 408.9 m^2^/g to 632.3 m^2^/g. The band gap energy of cellulose acetate@Ti-MIL-NH_2_ (2.42 eV) was slightly lower than pure Ti-MIL-NH_2_ (2.51 eV). The photocatalytic degradation of paracetamol (30 mg/L) in the presence of cellulose acetate@Ti-MIL-NH_2_ achieved 78% after 30 min under visible light irradiation (12 W).	[280]

## Data Availability

Not applicable.

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
