# Peer review of "Recent Progress on Tailoring the Biomass-Derived Cellulose Hybrid Composite Photocatalysts"

_polymers, 2022, doi:10.3390/polym14235244_

Round 1

Reviewer 1 Report

The manuscript "Recent Progress on Tailoring the Biomass-derived Cellulose Hybrid Composite Photocatalysts" by Yi Ding Chai , Yean Ling Pang * , Steven Lim , Woon Chan Chong , Chin Wei Lai , Ahmad Zuhairi Abdullah is acceptable. It deserves to be published after some moderate corrections.

The resolution of each figure should be increased, especially in Figure 3. The word and structure of chitosan/chitin are blurred and not clear.

Conclusion: Describe the study's limitations and future improvement in applying this type of material. 

The manuscript can be accepted after MINOR revisions are made, and some issues need to be amended, revised, and fixed before acceptance.

Author Response

The corrections/ rebuttal has been attached. 

Reviewer 2 Report

The potential applications of biomass-derived carbon-based photocatalytic materials for environmental remediation using the visible spectral region is an attractive and interesting topic. The dwindling supply of conventional fuels and the search for alternative raw materials for chemical production has made biomass an attractive resource that has significant potential to produce chemicals, fuels, and materials, paving the way for a sustainable future. Lignin/Chitin is a major fraction of biomass besides cellulose and hemicellulose that accounts for 40% of the total lignocellulosic biomass energy. However, little attention has been paid to the valorization of lignin due to its complex nature. Therefore, tweaking the materials by doping and constructing heterojunctions with biomass-derived carbon materials is effective and practiced in this area. The authors have aimed to review composite materials for photocatalysis applications.

The comprehensive review is written OK but the main concern of this review article and the areas that require clarification are given below.

1.       Recently, there was a review dedicated exclusively to biomass-derived materials for photocatalysis application published in RSC Adv. by B T Song et al 10.1039/D1RA05079F. Being this is the case, then what is the merit of the current review?

2.       In the section introduction, the fundamental reactions involved in photocatalysis, and the catalyst must be included.

3.       In Section 2.2; activated carbons are also obtained using the precursors such as Mango seed, Grape marc, and Wheat straw which has been recently published by Wickramaarachchi et al. Activated carbon is not only used for wastewater treatment but has also been popularly used as electrodes for energy storage (supercapacitor) applications. Please elaborate on this section while including the state-of-art other applications like batteries and supercaps.

4.       How do the synthesis conditions influence photocatalytic activity? Is it through the physicochemical properties of the material which has been altered through synthesis or due to the intrinsic nature of the biomass?

5.       In Section 2.4, the role of Chitosan/Chitin and its cross-linking nature with a reference to their morphology, particle size, and stability need to be detailed. Ramya Ramkumar et al have widely published the biopolymer chitin and its properties, please update the section.

6.       As section 3.1 is termed as “plant”, maybe chitin can be termed as “animal-derived chitin” for the sake of consistency.

7.       Are bacterial cellulose means microorganisms/fungi?

8.       The different synthesis routes reported for nanomaterials described in section 5 can be tabulated.

9.       Section 6.1 what is the role of metal doping in photocatalysis, and how it influences the bandgap?

10.   The key paper on MnO2 under Visible- Light Irradiation (https://doi.org/10.1002/slct.201600867) can be used for benchmarking the reported nanomaterials.

11.   Please make a link between biomass-derived cellulose to nanomaterials described in the later section. 

Author Response

The corrections/ rebuttal has been attached. Please see the attachment.

Round 2

Reviewer 2 Report

I have read the author's responses to my comments raised earlier. It appears that the responses are valid and the revised portion of the manuscript is well-framed. Therefore, to my opinion, the revised version is suitable to publish.